# Breaking the fast: first report of dives and ingestion events in molting southern elephant seals

Laura M. Charlanne [1✉], Laureline Chaise [2], Damien Sornette [2], Erwan Piot[3,4], Dominic J. McCafferty [5], André Ancel[1] & Caroline Gilbert[4,6]

Southern elephant seals (SES) experience a 'catastrophic molt', a costly event characterized by the renewal of both hair and epidermis that requires high peripheral vascular circulation. Molting animals are therefore constrained by high metabolic heat loss and are thought to fast and remain on land. To examine the ability of individuals to balance the energetic constraints of molting on land we investigate the stomach temperature and movement patterns of molting female SES. We find that 79% of females swam and 61% ingested water or prey items, despite the cost of cold-water exposure while molting. This behavior was related to periods of warm and low wind conditions, and females that dived and ingested more often, lost less body mass. We conclude that the paradigm of fasting during the molt in this species, and the fitness consequences of this behavior should be reconsidered, especially in the context of a changing climate.

[1]Université de Strasbourg, CNRS, IPHC UMR 7178, F-67000 Strasbourg, France. [2]Hex·Data, 847 Route de Frans, 69400 Villefranche-sur-Saône, France. [3]CNRS UMR5536, Université de Bordeaux, 33076 Bordeaux, France. [4]UMR 7179, CNRS/MNHN, Laboratoire MECADEV, 1 avenue du petit château, 91400 Brunoy, France. [5]Scottish Centre for Ecology and the Natural Environment, School of Biodiversity, One Health and Veterinary Medicine, College of Medical Veterinary and Life Sciences, University of Glasgow, Glasgow, UK. [6]Ecole Nationale Vétérinaire d'Alfort, 7 avenue du Général de Gaulle, 94704 Maisons-Alfort, France. ✉email: laura.charlanne@iphc.cnrs.fr

Molting is an important physiological event of partial or total renewal of integuments, which is influenced by season, growth, age, and environmental conditions[1,2]. This process occurs in a large variety of species, including mammals[3–5], birds[5,6], reptiles[7,8], fishes[9], and arthropods[10]. Molting is involved with a change in coat or plumage color, that may be related to reproductive success and camouflage in the environment[5,11], growth[10], thermoregulation[12], and defense against parasites or pathogens[7], hence playing an important role in the fitness of an individual. However, molt is an energy demanding process[13] requiring high peripheral vascular circulation to promote skin cell proliferation and tissue growth[14], and a high metabolic rate to balance heat loss[15,16].

Most Antarctic and sub-Antarctic phocid seals undergo an annual molt[4]. While seals are well adapted for an aquatic life, they are obliged to come ashore (on land or ice floe) to molt[17,18]. In phocids, regrowth of skin and fur requires a skin temperature above 17 °C[19]. Maintenance of this warm temperature in polar regions leads to elevated metabolic rate[15,16,20]. To reduce energy expenditure during the molt, seals are insulated with thick subcutaneous blubber[21,22], regulate peripheral blood flow, and avoid even higher heat loss by hauling out to minimize time in cold water, where heat loss is 25 times faster than in air[23,24]. Even with these actions, molting metabolic rate is 2–3 higher than resting metabolic rate[14,24]. The extent to which time in water, particularly during the molt, may affect thermoregulatory costs is poorly understood.

Southern elephant seals (SES) are one of only four species of phocids that undergo a "catastrophic molt", during which they renew both their hair and cornified epidermis[25,26]. Of the four species, they are the only one to undergo the molt in a polar environment (other species: northern elephant seal (NES), *Mirounga angustirostris*[22], Hawaiian monk seal (*Monachus schauinslandi*)[27], and Mediterranean monk seal (*Monachus monachus*)[28]). During this period, like other phocids, they lose a large proportion of their body mass (23% SES[29], 25%, NES[22], 17%, gray seals *Halicheorus grypus*[30]). This large change, combined with the increased peripheral circulation which would result in high heat loss in water, led to the conclusion that elephant seals fast and remain on land during the molt[22,31–33]. While ashore, they are exposed to a range of meteorological conditions. SES molt on subantarctic islands, they experience cold, wet, and windy conditions during the approximately 1-month haulout[34]. Environmental conditions strongly influence heat flux between an animal and the environment. SES minimize heat loss through social thermoregulation in large aggregations and through choice of different beach habitats such as grass or mud pools[35,36]. Therefore, we would predict that only animals with sufficient energy reserves to cover the cost of fasting, molting, and cold-water contact would be able to go to sea during the molting period.

A high energetic cost during the molt increases the depletion of energy reserves in blubber, with possible fitness consequences, such as lower survival or reproductive success for seals[37,38]. Climate in the Southern Ocean is rapidly changing, and environmental conditions are known to impact heat loss and molting phenology in phocids[15,39]. In this context, understanding how energetic constraints and individual strategies may help reduce metabolic costs of animals during key stages of their lifecycle is important. While it is commonly agreed that molting SES remain on land and rely on their reserves, breaking the fast has already been suggested as a mechanism to mitigate physiological costs while molting by restoring blubber reserves or limiting metabolic water production[31]. However, few studies examined movement and potential drinking or feeding behavior in this species throughout the molt. One way to investigate at-sea feeding or dinking behavior is to combine stomach temperature measurements with time-depth recorders (TDRs). This allows detection of possible cold water or prey item ingestion through stomach temperature changes and to record the depth and location of ingestion events. This approach has previously been used in seabird research (penguins, cormorants, albatrosses[40–43], and captive or non-captive pinnipeds (NES and Californian sea lions[20,44]), but not during the molt.

In this study, we investigated the behavior of adult female SES during the molt. Females were instrumented with TDRs to document diving behavior and stomach temperature pills to document ingestion events. We hypothesized that female SES would remain on land, either in mud pools or on the beach, fasting, at least in the early stages of the molt to minimize heat loss. If seals did spend time at sea, we investigated if weather conditions influenced this behavior. Last we hypothesized that only females in good body condition would spend time at sea, as they have sufficient energy reserves to compensate for high heat loss.

## Results

Combining diving analysis and stomach temperature recording, the monitoring duration was 7.0 ± 2.9 days of analyzed data (between 2 and 14 days of recording). Females arrived on land with a body mass of 314.7 ± 41.4 kg (BMI of 59.4 ± 5.1 kg/m²). Molt stage when animals were equipped was 54 ± 25% with 41% of the females caught at the initial stage ($n = 16$), 41% at the mid stage ($n = 16$), and 18% at the final stage of molt ($n = 7$). Body mass at the end of the monitoring averaged 294.5 ± 37.7 kg, resulting in a daily body mass loss of 3.2 ± 1.4 kg per day. At this point all females were at the final stage of molt, with more than 90% of hair renewed. During the monitoring period, the mean daily wind speed was 8.8 ± 2.7 m/s (ranging from 2.8 to 18.4 m/s), mean daily air temperature was 8.4 ± 2.4 °C (ranging from 4.4 to 15.3 °C), mean daily total sunshine duration was 355.3 ± 219.9 min (ranging from 0 to 855 min) and mean daily relative humidity was 70.2 ± 7.7% (ranging from 50.7 to 91.6%).

**Diving and surface swimming behavior**. Between 2014 and 2022, 39 female elephant seals were captured and monitored and 79% went to sea ($n = 31$). A total of 660 dives were recorded, grouped in 77 dive cycles (Table 1), for 39% of the monitored females ($n = 15$). Mean dive duration was 7.7 ± 3.6 min, with a mean time to reach maximum depth of 3.2 ± 3.1 min, followed by a mean surface phase duration of 2.2 ± 2.5 min. Dive cycles had an average duration of 1.4 ± 1.7 h.

We detected a total of 375 surface swimming events with an average duration of 1.8 ± 2.6 h grouped in 226 surface swimming cycles. Seventy-seven percent of monitored females ($n = 30$) performed surface swimming. Fourteen animals dived and surface swam. One female only dived.

On average, instrumented females spent time in the water (diving or surface swimming) every other day (% of monitored days with at-sea events = 49.2 ± 23.9%). The amount of time they spent in the water ranged from 9.0 to 92.8% of the day.

At-sea behavior was more diurnal, as 57% of the dives and 66% of the surface swimming occurred during the daytime.

**Stomach temperature**. Excluding temperature drops during the ingestions, mean at-sea stomach temperature was higher than on land (at sea = 37.1 ± 0.3 °C, on land = 36.5 ± 0.2 °C, Student's test, $t = -7.3$, df = 49.6, $p$ value < 0.0001). On land temperature analysis revealed hyperthermia events (from 1 to 3 events per individual) for 61.5% of the animals ($n = 24$), with a mean stomach temperature of 38.1 ± 0.3 °C. In total, 33 hyperthermia

events of $110 \pm 39$ min were analyzed, and the first at-sea event occurred within $24.5 \pm 21.0$ h after.

A summary of ingestion event variables is available in Supplementary Table S1. We recorded a total of 87 ingestion events (Table 2), from 80% of individuals ($n = 31$ from total $n = 39$), with an average of $2.8 \pm 2.1$ ingestions per individual. Among these events, 70% occurred during the daytime.

A total of 10 ingestion events were detected during a dive (12%) and 42 events occurred during surface swimming (49%). Due to bad Argos locations, the remaining 37 events (39%) were not clearly characterized as at sea or on land, but were not associated with complete submersion of the animal according to wet and light sensors. These ingestion events could have happened strictly on land or in the tidal zone.

Results of the different methods from the literature[44] are reported in Table 2, and based on the different methods, 25–53% of the ingestions were classed as potential prey ingestions.

**Determinants of female SES behavior**. At-sea behavior was not determined by initial physiological parameters; individual swimming score and the number of ingestions per female were influenced neither by BMI nor molt stage (Tables 3 and 4).

Environmental conditions played a role in SES behavior. We found a positive effect of temperature and a negative effect of wind speed on the probability of surface swimming (Table 5). Females tended to do more surface swimming during days of high temperature and days of low wind speed (Fig. 1A, B). The

probability of diving was not related to meteorological conditions (Table 5).

We also found a positive effect of sunshine duration on the probability of ingestion. The probability of recording individuals that ingested increased during days with high total sunshine duration (Fig. 1C). The effect of relative humidity and wind speed on ingestion was not significant (Table 5).

**Consequences on female SES condition**. Female SES condition at the end of the molt was influenced by their behavior. For females going to sea, we found a positive relationship with the swimming score (PC1) and the number of ingestions on daily body mass loss (Table 6). Females that dived and ingested more often were in better body condition at the end of the molt (Fig. 2A, B).

**Discussion**. Through the use of TDRs and STP, we identified surface swimming, diving, and ingestion events during the molt of female SES. To our knowledge, this study is the first to show evidence of SES breaking the fast during this energetically costly molt period.

We found no evidence of adverse effects of the STP on the behavior and body condition of animals at the end of the monitoring period. The STP assembly remains in the stomach, and the potential effect on satiety or ability to ingest was not possible to test. However, no such adverse effects have been described in experimental studies on pinnipeds[44], probably due to

---

**Table 1 Summary of diving parameters from 31 female southern elephant seals during the molt.**

|  | Mean (SD) | Min.–max. |
|---|---|---|
| Estimated time spent at sea per day per individual (min) | 109.32 (98.3) | 1.0–391.0 |
| Mean number of at-sea events per day per individual | 1.1 (0.8) | 0.1–2.8 |
| Total number of dive cycles per individual | 5.1 (5.5) | 1–22 |
| Total number of dives per dive cycle | 8.6 (10.8) | 1–49 |
| Total number of dives per individual | 44.0 (55.0) | 1–157 |
| Maximum dive depth (m) | 12.5 (7.3) | 5–43 |
| Dive duration (min) | 7.7 (3.6) | 0.2–22.5 |
| Time at surface (min) | 2.2 (2.5) | 0.3–19.3 |
| Surface swimming duration (h) | 1.8 (2.6) | 0.2–26.7 |

Among 31 female who went at sea, 15 dived ($n = 660$ dives, 77 dive cycles, and 226 episodes of surface swimming; 2014–2022). "At-sea events" include both surface swimming and diving.

---

**Table 2 Summary of ingestion type based on Kuhn and Costa (2006) results, from 31 female southern elephant seals monitored between 2014 and 2022 ($n = 87$ events).**

| Method | Threshold for feeding events | Number of feeding events |
|---|---|---|
| Index of rate of temperature recovery $I$ (s. °C$^{-1}$) | $I$ value > 250 s °C$^{-1}$ | $n = 39$ (44%) |
| Area above the curve (s °C) | Integral > 3000 s °C | $n = 25$ (29%) |
| Time of temperature recovery ($\Delta t$ between $T_2$ and $T_3$) | $t_{rec} > 35$ min | $n = 46$ (53%) |
| Delta temperature $T_1 - T_2$ | $\Delta T > 4.7$ °C | $n = 22$ (25%) |

$T_1$ is the initial stomach temperature and $T_2$ the minimum temperature during the ingestion event.

---

**Table 3 Best models to explain female southern elephant seals swimming score variability.**

| Response variable | Behavior/group | Explanatory variables | $X^2$ | Df | $p$ | AICc |
|---|---|---|---|---|---|---|
| PC1 | Diving | Monitoring duration | 13.43 | 1 | <0.001*** | 63.3 |
| PC2 | Surface swimming | Monitoring duration | 0.34 | 1 | 0.56 | 56.5 |

ANOVA table of the best linear mixed effect models for at-sea behavior (PC1 and PC2): diving ($n = 15$) or surface swimming ($n = 31$).
***p value < 0.001.

**Table 4 Best models to explain female southern elephant seals ingestion behavior variability.**

| Response variable | Location | Explanatory variables | Coefficient ± SE | Z | p |
|---|---|---|---|---|---|
| Number of ingestions | At sea | Intercept | 0.54 ± 0.43 | 0.95 | 0.20 |
| | | Monitoring duration | 0.03 ± 0.05 | 1.07 | 0.51 |
| | On land | Intercept | −1.95 ± 2.95 | −0.66 | 0.51 |
| | | Monitoring duration | 0.29 ± 0.48 | 0.60 | 0.55 |

Summary of coefficients and goodness-of-fit indices from the best models for the number of ingestions (generalized linear models fitted with negative binomial law), at sea ($n = 32$), or on land ($n = 7$).

**Table 5 Summary of coefficients and goodness-of-fit indices from the best models for environmental conditions effect on female southern elephant seals behavior.**

| Response variable | Explanatory variables | Coefficient ± SE | t | p |
|---|---|---|---|---|
| Daily proportion of diving females | Intercept | −1.7 ± 0.8 | −2.0 | 0.04* |
| | Temperature | 0.06 ± 0.08 | 0.8 | 0.44 |
| | Wind speed | −0.09 ± 0.06 | −1.4 | 0.16 |
| Daily proportion of surface swimming females | Intercept | −1.6 ± 0.6 | −2.7 | 0.008** |
| | Temperature | 0.21 ± 0.06 | 3.7 | <0.001*** |
| | Wind speed | −0.10 ± 0.05 | −2.1 | 0.03* |
| Daily proportion of ingesting females | Intercept | −3.7 ± 1.7 | −2.1 | 0.03* |
| | Sunlight | 0.001 ± 0.0006 | 2.8 | 0.02* |
| | Humidity | 0.03 ± 0.02 | 1.7 | 0.09 |
| | Wind speed | −0.09 ± 0.05 | −1.8 | 0.07 |

Generalized linear models were fitted with binomial (probability of ingesting) or quasibinomial (probability of diving and surface swimming) law. Females behavior is separated as diving behavior ($n = 15$ females), swimming behavior ($n = 31$ females), and ingesting behavior ($n = 31$ females).
*$p$ value < 0.05; **$p$ value < 0.01; ***$p$ value < 0.001.

the short retention time and small size of the STP compared to the volume of the stomach.

Our data revealed that female SES went to sea during the molt, as reported in Table 1 and Fig. S1. Indeed, nearly half of equipped individuals performed several dives, from 5 to 43 m.

To our knowledge, this behavior has not been reported yet in a species displaying a "catastrophic" molt, including aquatic birds and mammals[4]. In marine mammals, pinnipeds undergoing such an extreme physiological event, such as elephant seals[22,31–33,45], have always been thought to rely exclusively on energy reserves in blubber during the molt. All birds experiencing catastrophic molt are known to fast and avoid cold water contact, as demonstrated in cormorants[46] and penguins[47]. Few diving birds are known to break the fast and dive during the molting period, but they undergo a rapid simultaneous molt that is not reported as "catastrophic" (common eider *Somateria mollissima*[48], lesser snow geese *Chen caerulescens caerulescens*[49], and grebes *Podiceps* sp.[50]). Therefore, our results reveal that SES may be the first recorded case of a species experiencing a catastrophic molt while also spending time at sea, despite the potential energetic costs.

Although female SES dove during their molting fast, diving parameters differ from diving foraging parameters. Indeed, most of the dives were not deeper than 11.5 m and lasted on average for almost 7 min, compared to average foraging dive depth and durations of 400 m and 30 min[51–54], respectively. SES are known to perform deeper dives according to prey distribution, migration, or predation avoidance[55–57]. Cold water exposure at shallow depths during this period thus requires further investigations to better understand the reasons for such behavior.

According to our data, female SES dive during the molt but also ingest water and/or food, which challenges the paradigm of fasting. Indeed, stomach temperature clearly shows that 80% of diving females ingest water and/or prey. Such behavior has already been suggested[31] as metabolic rate calculations based on water flux during the molting period indicate that water ingestion takes place, but no food was found in the stomachs of females. In our study, the type of ingestion (water and/or prey) remains unclear. Indeed, the results differ between the different methods of estimation, as supposed feeding events represent approximately 25–53% of ingestion events (Table 2), which is greater than previously described through behavioral and movement patterns[58].

It is difficult to distinguish between water or prey ingestion events in pinnipeds. While index I values (index of the rate of stomach temperature recovery, see Method section) of less than 30 s °C⁻¹ always denoted water ingestion for albatrosses[23], for pinnipeds, this distinction was more complicated[44]. In a previous experimental study, index I values for prey ingestion ranged from 55.1 to 4380.0 s °C⁻¹, while water ingestion ranged from 37.8 to 764.0 s °C⁻¹. The previous study used a threshold of 250 s °C⁻¹ to

distinguish fish from water consumption, as this value lowered the error rate of identification, but uncertainty remained[44]. We used the same threshold in our study but as sea surface temperature was different (around 14 °C in the experimental study[44], and 8 °C in Kerguelen during the molting period[34]), it may distort variable calculations and so ingestion type. In our study, as 49% of ingestion events occurred during surface swimming (i.e., depth between 0 and 5 m), and regarding prey repartition[56], half of ingestion events may likely involve water.

Across taxa, catastrophic molts in birds and mammals have always been associated with fasting[4]. It appears that SES may be the first species reported to break the fast during this energetically costly period, but why?

In this study, we found that meteorological conditions influenced the behavior of molting female SES. The weather effect on social thermoregulation and habitat preference has already been described during the molt in this species[35,59]. Meteorological conditions influence the behavior of molting pinnipeds, such as haulout frequency and time spent on land, but it has only been investigated in species of phocid seals that do not undergo a catastrophic molt and keep foraging at sea during this period[15,39]. Our data revealed that during days with low wind speed and days with high air temperature, female SES tend to swim more often. The same parameters were reported to influence the number of molting Weddell seals (*Leptonychotes weddellii*) hauling out on ice, but surprisingly, it was the opposite behavior, as Weddell seals tend to remain in the water when wind speed is elevated and air temperature are low[39]. Meteorological conditions strongly differ between Kerguelen and Antarctica, where Weddell seals are found. If daily wind speed is nearly equal (8.8 ± 2.7 m/s, ranging from 2.8 to 18.4 m/s in Kerguelen and 5.3 ± 4.2 m/s, ranging from 0.7 to 19.8 m/s in Syowa station), air temperature does not go below 0 °C during the molting period in Kerguelen (8.4 ± 2.4 °C, 4.4–15.3 °C versus −8.3 ± 5.3 °C, with a maximum of 1.7 °C in Syowa). Weddell seals and SES are not exposed to the same environmental conditions during their molting period, but both could lead to thermal discomfort. The thermoneutral zone (TNZ) is a range of temperature between a lower and upper threshold, between which the metabolic cost associated with thermoregulation is the lowest[60]. Out of this range, animals are hypothermic or hyperthermic, resulting in

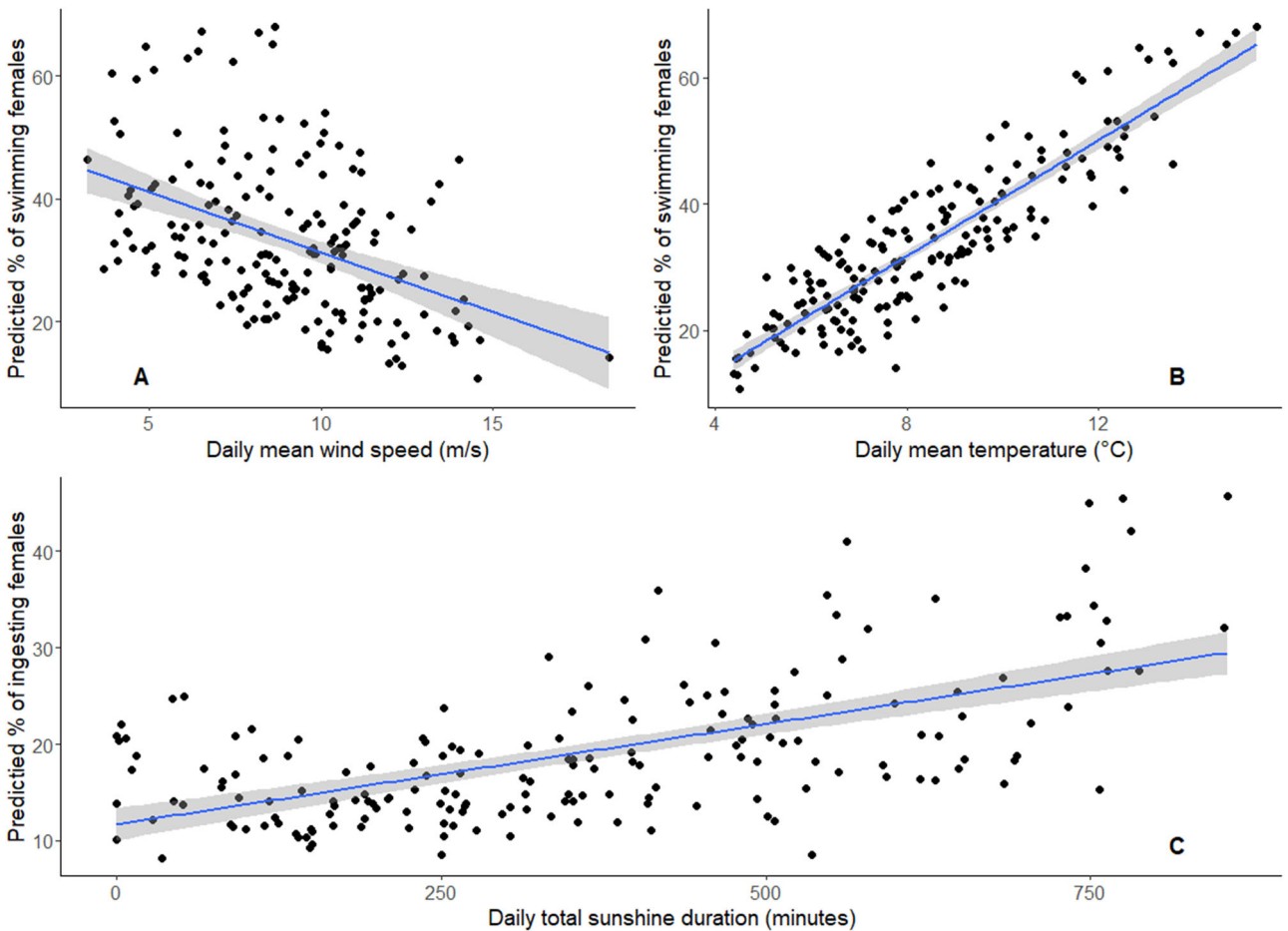

**Fig. 1 Relationships between meteorological conditions and behavior of female southern elephant seals using. A** GLM predicted daily proportion of swimming females with daily mean wind speed (m/s). **B** GLM predicted daily proportion of swimming females with daily mean temperature (°C). **C** GLM predicted daily proportion of females that ingest with total daily sunshine duration (min/day).

**Table 6 ANOVA table of the best linear mixed effect models for female southern elephant seals condition at the end of the molting period.**

| Response variable | Location | AICc | Explanatory variables | $X^2$ | Df | p |
|---|---|---|---|---|---|---|
| Body mass loss | On land | 46.9 | Mass start | 0.38 | 1 | 0.53 |
| | | | Length | 0.27 | 1 | 0.61 |
| | | | Monitoring duration | 1.41 | 1 | 0.24 |
| | At sea | 126.7 | Mass start | 0.02 | 1 | 0.89 |
| | | | Length | 0.37 | 1 | 0.54 |
| | | | PC1 | 8.19 | 1 | 0.004** |
| | | | Number of ingestions | 4.42 | 1 | 0.03* |
| | | | Monitoring duration | 10.65 | 1 | 0.001** |

Body mass loss is used as a proxy for female southern elephant seals condition. Females are grouped by location: on land ($n = 6$) or at sea ($n = 12$).
*$p$ value < 0.05; **$p$ value < 0.01.

higher metabolic costs (thermogenesis) or heat dissipation (thermolysis) to avoid deleterious effects of overheating on the organism, such as water loss, cell-damage, and disruption of reproductive function[61,62]. Consequently, any supplementary energetic cost required for thermogenesis or thermolysis in both hypothermia or hyperthermia may increase energy depletion[37,38].

Moreover, as phocid seals obtain their water requirements from the metabolism of fat reserves[63,64], compensating for metabolic water loss triggered by heat stress could be extremely costly. Going to sea could be a way to avoid extreme weather events when seals are out with their TNZ, leading to thermal discomfort and to other physiological effects. As SES remain in a warmer environment, they should avoid overheating and minimize elevated metabolism and increased water loss on warmer days, while Weddell seals, exposed to colder conditions, would go to sea to reduce metabolic costs of thermogenesis in bad weather conditions.

We were not able to obtain solar radiation in W/m², but instead used sunshine duration (min per day), which is a less precise estimate of the thermal power received at the surface of the skin. The effect of sunshine duration was not significant in initiating at-sea movements in our females, but solar radiation was already reported to be positively correlated with swimming behavior during the breeding period of females NES[65] and other pinnipeds[66,67]. Heat dissipation is maximized by peripheral vasodilatation and by an important thermal gradient through thermal windows (surface temperature-air temperature[68]). If peripheral circulation is already increased during the molt, and solar radiation heats blood at the skin surface, heat loss by radiation or convection may not be sufficient to cool down and prevent thermal stress during warm days. Because of the high cooling capacity of water, going to sea would be the most efficient method to dissipate heat[66]. This suggests that SES could be subject to thermal stress and go into the water to avoid

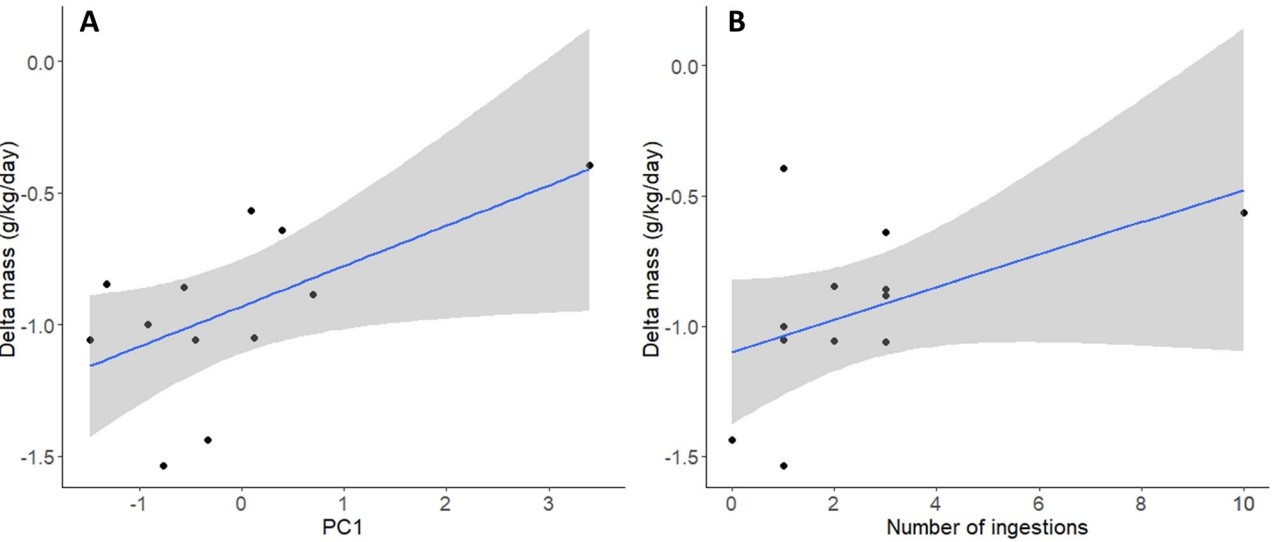

**Fig. 2 Daily mass loss per unit of body mass (g/kg/day) of female southern elephant seals according to their behavior. A** Diving score (PC1 as individual values on the first component of the PCA) and **B** number of ingestions per individual. Individuals with high PC1 values perform more dives, last longer, and show more surface swimming.

overheating, especially during the molt when vasodilatation at the skin surface occurs.

Total sunshine duration was correlated with the number of ingestion events in our females. As ingestion events were rare during a dive (12% of ingestion events), this would be more consistent with drinking than foraging. Drinking at sea has already been reported in gray seals[67] and would be in line with a behavioral response to thermal stress to avoid supplementary metabolic costs and blubber depletion needed to restore hydration if they remain on land.

Although the majority of our results confirm a behavioral response to thermal stress, few hyperthermia events were detected (from 1 to 3 events for 61.5% of the animals) and were not related to an immediate at-sea event (first at sea event within 24.5 ± 21.0 h after a detected hyperthermia). In breeding NES, a strong effect of solar radiation on surface temperature was reported and was related to at sea movement[65]. As heat is gained by radiation, an increase in surface temperature may be most relevant to hyperthermia events. Considering the blubber thickness in phocid seals[69–71], behavioral responses to high surface temperature (i.e., going to sea) may occur rapidly after an increase in surface temperature, probably before stomach temperature increased through heat diffusion by conduction. This may question the relevance of using stomach temperature for detecting overheating in response to environmental factors in this species.

Finally, our data did not show any link between at-sea movements and stage of molt, where we expected more at sea movements when heat loss is reduced at the end of the molt[15]. This may indicate that heat loss and the metabolic cost of cold-water immersion are not limiting factors for this behavior, in line with possible thermal stress events and the need for heat dissipation rather than conservation.

Considering high energetic constraints triggered by cold water contact and previous studies reporting that body condition could influence molting duration[72], we thought female SES with a high BMI would be less constrained by energetic trade-off and spend more time at sea than females in lower body condition. However, our data did not find any evidence that body condition influenced the behavior of molting females, neither at-sea movements nor ingestion events. Molt stage also did not influence female behavior despite less energetic constraints at the end of the

molt[15], which suggests an environmental rather than a physiological driver of this behavior.

Moreover, going to sea during the molt might not be as costly as we believed. Daily specific body mass loss of our females was a good indicator of energetic loss during the molt. A previous study reported that the mass loss of female elephant seals averages 4.7 kg/day, during 25–30 days of molting[31]. Surprisingly, females that went to sea did not lose more body mass than females remaining on land but had higher stomach temperatures, which could reflect a higher metabolic rate, hence higher energy loss at sea. However, previous studies on molting Arctic seals (ringed *Pusa hispida*, spotted *Phoca largha*, and bearded seals *Erignathus barbatus*) reported that metabolic rates were similar when resting in air and water[16]. An increase in stomach temperature might be detected at the onset of diving as a result of heat generated by muscles during swimming[73], which questions the reliability of STP to estimate metabolic variation.

According to our results, females may benefit from trips to sea. Among the females that went to sea, the more they dived and ingested, the less mass they lost. The diving profile of these females was very similar to foraging diving profiles, suggesting they could ingest prey items during their trip. Although depth and duration are very different from typical foraging dives, high shallow biomass sites related to phytoplankton bloom have been located in the vicinity of the Kerguelen Plateau during spring and summer, with a peak between December-February[74]. Many studies have shown that juvenile SES perform shallower dives during their first year at sea and that their diet is mainly composed of krill (*Euphausia* sp.)[75–77]. As molting females are constrained to stay close to the island until the end of the molting process, they could be limited by the bathymetry and benefit from shallower prey such as krill, which could restore body reserves if the molting period is too long or too costly to rely on body fat. As these are very small prey items, that could also explain the difficulty of distinguishing between water and prey items from ingestion curves. However, no precise information on what was ingested could be determined with this method, and thus this question requires further investigation. Other methods such as sonar tag or video camera[78–80] could collect further important information which would determine precisely what is ingested during these trips to sea during the molt.

Energetic constraints and metabolic costs associated with molting behavior, as well as the possible fitness consequences, still remain unclear.

Whether seals remain on land to limit heat loss and metabolic costs from unavoidable perfusion during the molt or whether this behavior allows a faster molt remains uncertain. Avoiding cold water would limit energy expenditure and blubber depletion, while a faster molt would limit fasting duration and extend the next foraging period[72], resulting in higher at-sea body condition, survival, and reproductive success[37,38]. Previous studies on ringed and spotted seals did not report supplementary energy expenditure at sea[16], and our data did not show any difference in body mass loss between at-sea and on-land females. This highlights that the physiological cost of molting, i.e., for tissue regeneration, may be greater than the cost of heat loss in cold water. Spending an extended time at sea would be costly by slowing down the molt. Unfortunately, because the molting stage is difficult to determine and to assess without regular visual checks, we were not able to investigate to what extent the amount of time spent at sea influenced the duration of the molting period. This would be of strong interest as this equilibrium of haul out frequency and molt duration may be compromised by current global changes and rapid warming at the poles[81]. Thermal stress could affect the behavior of seals, increasing molt duration and causing a greater depletion of blubber reserves. In this context, knowing how molt duration impacted ingestion behavior in females is of interest. Seals with a slower molt may need to restore their body reserves to make sure they end the molting process with sufficient body condition to ensure good foraging success and survival during their next post-molt foraging trip. Thus, the link between energetic constraints, molting phenology, and behavior of SES on land needs further investigation.

This study found that female SES travel at sea during the molt, performing shallow and short dives or surface swimming, and these behaviors were associated with ingestion events. This shows that SES frequently break the fast and may not always remain on land, relying exclusively on blubber reserves. While environmental factors responsible for heat stress were determinants of swimming behavior and likely water ingestion, physiological and fitness consequences of diving and ingestion events need to be further investigated. This will allow a better understanding of the energetic constraints that drive this critical period and the extent to which thermal stress may impact SES. Considering rapid changes in polar climates and possible fitness consequences, the effect of warmer conditions is of particular interest for the conservation of polar phocids.

## Methods

**Data collection and analysis**. All scientific procedures were approved by the Ethics Committee of the French Polar Institute. Data were collected during the austral summer molt (December-February) between 2014 and 2022. The field work took place at three colonies within the Kerguelen Archipelago: Pointe Suzanne (49°26'S, 70°26'E) between 2014 and 2019, Estacade (49°16'S, 70°32'E) in 2020 and 2022, and Port-Aux-Français (49°34'S, 70°21'E) in 2021.

Female elephant seals were captured during the molt and anesthetized using tiletamine and zolazepam (Zoletil® 100[59,82]). Individuals were equipped with devices provided by Wildlife Computers (USA): TDRs (TDR10-LX-340, 57 × 38 × 30 mm, 75 g in air), Argos satellite transmitters (SPLASH10-L-309D, 76 × 55 × 32 mm, 125 g in air) and Stomach Temperature Pills (TDR-STP-207D, 63 × 21 mm, 31 g in air). Argos location was obtained each day from resulting transmissions. Locations are classified as 3, 2, 1, 0 according to the error radius (<250 m,

250–500 m, 500–1500 m, >1500 m). We show an example of good at-sea locations in Supplementary Fig. S1. Stomach temperature pills (STP) were placed in the stomach under anesthesia using a lubricated flexible tube, and SPLASH10/TDR10 tags were glued on the head with epoxy bi-component glue (Araldite®). Data-logging tags were set to measure depth (resolution of 0.5 m with an accuracy of 1% of reading), light-level (range from $5 \times 10^{-12}$ to $5 \times 10^2$ W/cm² with a resolution of 20 units/decade), stomach temperature (resolution: ±0.1 °C; accuracy: ±0.3 °C), and wet/dry sensor every 10 s. Tags were retrieved at recapture, while remaining STP were naturally ejected later at sea or on land. Individuals were recaptured between the completion of the molt and their final departure to sea.

Of the 55 instrumented seals, we obtained stomach temperature data over 2 days from 39 of the seals. Data were analyzed starting from 6 h following capture (to discard the effects of anesthesia) until the STP was ejected or the seal was recaptured. For each individual, the molting stage was visually estimated at both capture and recapture and during daily monitoring (0–40%: initial stage; 50–80%: mid stage; >90%: final stage, as described in previous study[35]). Eighteen of the instrumented seals were weighed at capture and recapture, allowing us to calculate the daily mass loss (kg/day). Length (m) was calculated from the mean length measured at capture and recapture. From this, we calculated a body mass index (BMI, kg/m²).

Local meteorological data was retrieved from Météo-France base station archives in Port-Aux-Français (Kerguelen archipelago). These data included temperature under cover (°C), relative humidity (%), sunshine duration (min), and wind speed (m/s). Data were recorded per hour and summed per day for sunshine duration.

Data were processed and analyzed using "pandas," a Python data analysis package (versions: Python 3.10; pandas 1.5.0), and R software (R version 4.1.2).

**Diving pattern analysis**. Because Argos locations were poor, diving patterns were analyzed only from depth, light level, and wet/dry sensor data. When necessary, a depth-zero correction was made on land at the time of capture. Diving behavior was determined using a Python algorithm especially developed for this study (Hex·Data, France). To make sure light level variations involved depth and not daily changes, day-night light-level variations were removed using a seasonal decomposition function based on moving averages (function "seasonal decompose" from Python package "statsmodels"; version 0.13.2). We combined a depth threshold of 0.5 m and a "wet" display to spot the start of a dive. To distinguish mud pools (wallows) from sea-water baths, we made sure the light level varied by more than 5 units (values ranging from 0 to 255) during the presumed dive.

At-sea behavior was characterized by (1) a submerged phase, (2) a maximum depth, and (3) a surface phase. We distinguished surface swimming and diving behavior according to maximum depth. Surface swimming was defined to occur between 1 and 5 m, with a maximum surface time of 20 min between two submerged phases (based on our own data exploration, Fig. 3A). Dives were described by a descent, a bottom, and an ascent phase with a maximum depth greater than 5 m and showed different profiles (Fig. 3B–D). A dive cycle was characterized by at least two successive dives separated by less than 20 min at the surface, as surface duration intervals showed a break around 17 min. Events were described by their mean duration, mean and maximum depth, and frequency.

**Stomach temperature analysis**. Ingestion events were identified by a sharp decline in stomach temperature followed by a

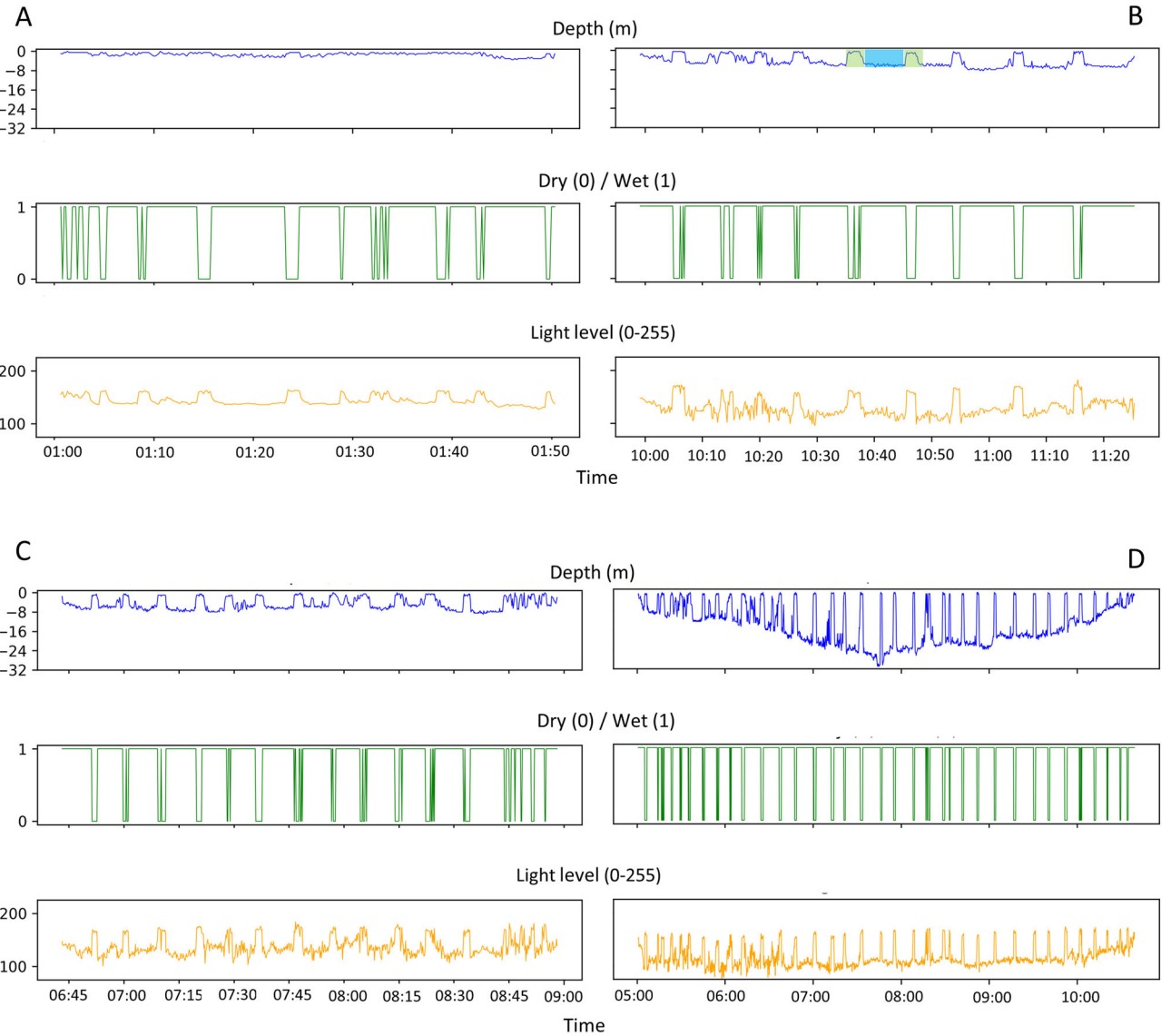

**Fig. 3 Diving parameters during different at-sea behaviors: diving or surface swimming.** Surface swimming occurs between 1 and 5 m, while a dive is characterized by a maximum depth of more than 5 m. A dive or a surface swimming cycle is characterized by at least two successive dives or surface swimming separated by less than 20 min at the surface. **A** represents a surface swimming cycle and different profiles of dive cycles (**B**-**D**) of a female southern elephant seal in 2017. Each dive is defined by the succession of a submerged phase (blue area) and a surface phase (green area) (**B**).

logarithmic increase (Fig. 4A–D). The ingestion detection algorithm identified:

1. The minimum temperature below 36.5 °C (under the lowest core temperature described for elephant seals[36,44]), referred to as $T_2$
2. The previous stable temperature before the drop ($T_1$, backward from $T_2$)
3. The point at which temperature returned to baseline for at least 10 min[44,83] ($T_3$, forward from $T_2$)

$T_1$, $T_2$, and $T_3$ are illustrated in Fig. 4A. Events that were close together were merged into a single event. Each ingestion event was assigned to a female behavior: on land, diving, surface swimming, or unknown. Following the algorithm analysis, a visual check of ingestion curves led to the removal of 12 anomalous events (non-logarithmic curves or abrupt variations consistent with STP measurement errors). The STP was considered lost when the stomach temperature reading remained constant for at least 3 h. Outliers were deleted using a filter based on the rolling average and bounded by the first and third quartiles.

To examine STP temperature drift over time, we downsampled the data to one measurement every 15 min (factor of 100) and examined the trend in temperature over time using Kendall's tau correlation coefficient. The temperature increased for 16 individuals, decreased for 12 individuals, and did not change for the remaining 11 individuals. The mean absolute value of the correlation coefficient was $\tau = 0.11 \pm 0.09$, with the mean absolute value of the slope estimator $s = 5.1 \times 10^{-7} \pm 6.2 \times 10^{-7}$, corresponding to a variation of ±0.1 °C every 10 days (for 72% of the 39 individuals; $n = 31$). Therefore, based on the temperature pill accuracy (±0.3 °C) and mean survey duration (over a week), we considered that the signal drift was trivial and chose not to apply a correction.

STP were synchronized with diving patterns, that were filtered with a seasonal decomposition of light-level function to cancel the effect of the day/night cycle. As this function required 48 h to work, six animals with less than 2 days of recording were

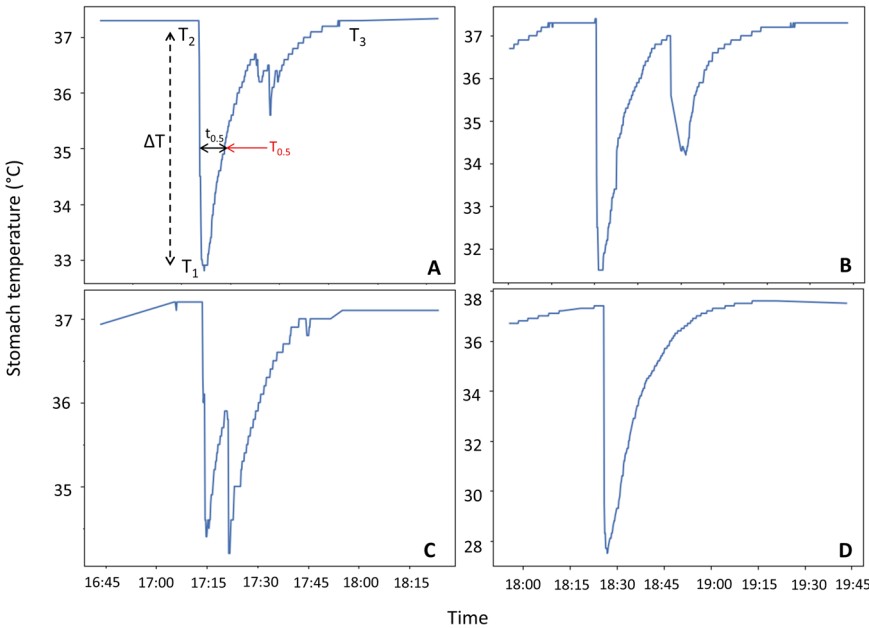

**Fig. 4 Stomach temperature variations (°C) during ingestion events recorded for two female Southern elephant seals in 2014 and 2019. A** Standard temperature variation. **B, C** Double ingestions. **D** Extreme temperature variations. $T_1$ is the initial temperature, $T_2$ the minimum temperature, $T_3$ the recovery temperature, $\Delta T$ the difference of temperature between $T_1$ and $T_2$, $T_{0.5}$ the half-way recovery temperature between $T_1$ and $T_2$ ($T_{0.5} = T_2 + \Delta T/2$) and $t_{0.5}$ the time of half-way recovery (black arrow: $t_{0.5} = t_{T0.5} - t_{T1}$). The area above the curve (integral) was calculated from the asymptote.

discarded (R documentation). Eight seals ejected the pill before recapture. In total, 80% of monitored seals were recaptured with the temperature pill still recording ($n = 31$), with an average retention duration of $7.2 \pm 2.9$ days.

To distinguish between water and prey ingestions, we followed two published methods[43,44], using the index of the rate of stomach temperature recovery ($I$, s °C$^{-1}$), calculated as follows:

$$I = t_{0.5}/(T_1 - T_2)$$

where $T_1$ (°C) is the initial stomach temperature before the drop due to the ingestion of sea water, $T_2$ (°C) is the minimum temperature reached during the drinking or feeding event, and $t_{0.5}$ (seconds) is the time from the beginning of the temperature decrease to the point of half recovery (half-way temperature between initial $T_1$ and minimum $T_2$; Fig. 4A). The threshold of 250 s °C$^{-1}$ was used to distinguish between fish and water consumption, as previously reported[44].

We also calculated the area above the curve (integral) of drop in stomach temperature during ingestion events, as it may be related to ingested mass in NES, based on the trapezoid method[44,73] (for more details, see Supplementary Fig. S2).

Ingestion events were compiled with dry/wet and light sensor to identify whether they occurred at sea or on land (mud pools).

We investigated on-land stomach temperatures to detect hyperthermia events that could have initiated at sea movements. We calculated the mean stomach temperature on land ($36.6 \pm 0.2$ °C) and considered hyperthermia if values were higher than the 97.5% quantile for more than 1 h.

**Statistics and reproducibility**. Statistical analyses were performed with R software (R version 4.3.1). Results are expressed as mean ± standard deviation (minimum to maximum), unless otherwise specified.

As at-sea behavior and diving parameters were variable among individuals, we defined a swimming score to determine the amount of time spent in water, including surface swimming and diving. To do so, we used a centered-scale Principal Component

Analysis (PCA) to account for diving parameters (number of dives per individual, number of surface swims per individual, and mean dive or surface swimming duration). The first component (PC1) accounted for 59.0% of the variation, and the second component (PC2) accounted for 23.8% of the variation. PC1 was characterized by the number of dives, the number of surface swims, and the mean dive duration. PC2 was defined by the surface swimming duration. We decided to define three groups according to the behavior of each female (diving, surface swimming, on land, Fig. 5) in the models when group size was sufficient. As an animal's behavior was well separated by the PCA, PC1 values were kept as a proxy of swimming score for the group Diving, and PC2 for the group Surface swimming.

To investigate the reason for variability in SES swimming score, we used linear mixed models (LMM) with molting stage when equipped, BMI, monitoring duration as explanatory variables, PC1 (group diving) or PC2 (group surface swimming) as response variables. To determine if physiological parameters influenced ingestion behavior, we ran a generalized linear mixed model with molting stage when instrumented, BMI, monitoring duration as explanatory variables, ingestion counts per individual as response variable. To correct for overdispersion, we used a negative binomial distribution[84]. Each time, animal ID was used as random. For each model, an ANOVA type II was performed (Anova function in R library "car").

Then, to determine whether weather influenced when seals spent time in the water we ran a generalized linear model with temperature, relative humidity, sunshine duration and wind speed as explanatory variables, and the daily probability of female SES diving, surface swimming, and ingesting as response variables. Models were fitted with a binomial distribution (logistic regressions for probabilities), checked for overdispersion, and adjusted with a quasibinomial distribution if necessary.

Last, to determine whether changes in SES behavior and molt phenology influenced their condition before their next travel, we used LMM with the swimming score, ingestion counts, body mass/length and molting stage at capture, and monitoring duration as explanatory variables, body mass loss (kg) as response variable, and animal ID as random.

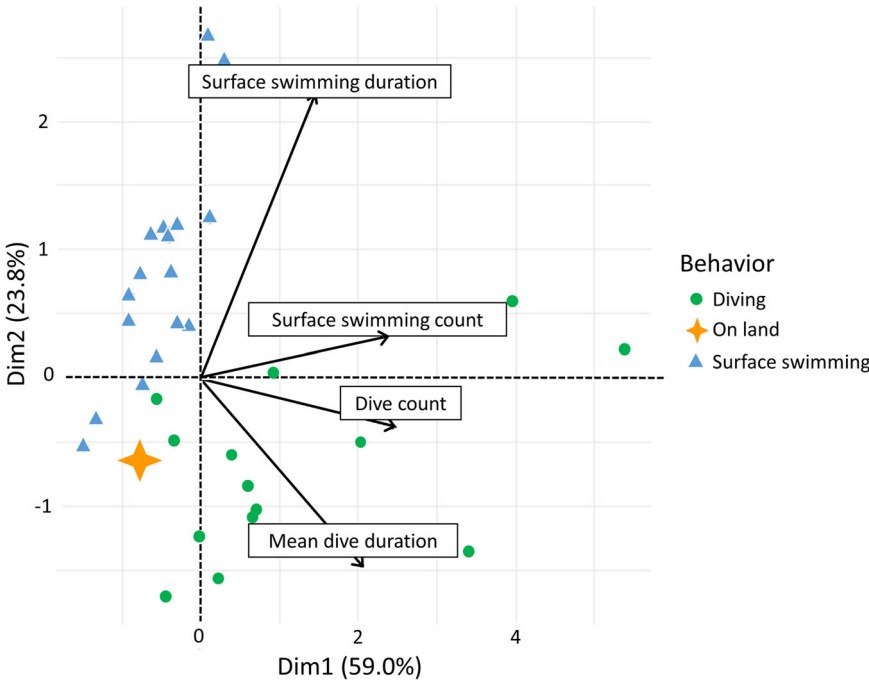

**Fig. 5 Principal Component Analysis (PCA) including number of dives per individual, number of surface swims per individual, and mean dive or surface swimming duration.** The first component (PC1) accounted for 59.0% of the variation and the second component (PC2) accounted for 23.8% of the variation. PC1 was characterized by number of dives, number of surface swims, and mean dive duration. PC2 was defined by surface swimming duration. Individuals are reported on the PCA and identified through their behavior (red circles, diving; orange star, on land; blue triangle, surface swimming only). Orange star represents eight individuals on the same point.

**Reporting summary**. Further information on research design is available in the Nature Portfolio Reporting Summary linked to this article.

## Data availability
All diving data are available in the figshare public data at https://doi.org/10.6084/m9.figshare.24773235.

## Code availability
The Python code is available on request from the authors. The R code can be downloaded from Zenodo[85].

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

## Acknowledgements

The overall study was funded by the Institut Polaire Français Paul-Emile Victor (IPEV Programs 1037 HEnergES and 1201 Cycleleph) and doctoral fellowships of French Ministry of Higher Education and Research. We thank the Terres Australes et Antarctiques Françaises for logistic support and the fieldwork volunteers who helped in tag deployments and retrievals (in alphabetic order; Hassen Allegue, Jérôme Badaut, Batshéva Bonnet, Antoine Bertault, Laureline Chaise, Laura Charlanne, Océane Cossu-Doye, Lucas Delalande, Liane Dupon, Susan Gallon, Manon Ghislain, Christophe Guinet, Adélie Krellenstein, Alex Lathuilliere, Marion Nebot, William Paterson, Julie Pauwels, Baptiste Picard, Erwan Piot, Manon Potin, Camille Toscani, Martin Tournier, Yakov Uzan, Coline Vulliet, Pauline Vuarin). D.J.M. was funded by the Carnegie Trust for the Universities of Scotland.

## Author contributions

C.G., A.A., and D.J.M. conceived and designed the overall study. L.C., L.M.C., and E.P. contributed to field work. L.C. and D.S. conceived the algorithm and performed the analyzes. L.M.C. performed the statistical analyzes and wrote the manuscript. All authors reviewed and agreed on the final content of the manuscript.

## Competing interests

The authors declare no competing interests.
