## [Peer Review File · Communications Biology]

Reviewers' comments:

Reviewer #1 (Remarks to the Author):

In this manuscript the authors report that some elephant seals spend time at sea, occasionally performing dives and feeding, during the molt. This is in contrast to the commonly held view that elephant seals remain on land and fast for the duration of the fast. While it is exciting that they have documented this with stomach temperature pills and data loggers, I do not think this manuscript is suitable for publication in *Communications Biology*. The authors need to provide more context on why this is important, should perform additional analysis, and improve the language so the reader can better understand the methods, results, and discussion. Due to vague language, not enough detail, and unclear writing it was difficult to understand all of the methods. Below I explain 3 major concerns followed by more detailed information about those concerns and language.

Major concerns:

My first major concern is that this paper is too elephant seal centric. The authors need to better set up the importance of molt in the intro. The paper would be much stronger if more background was provided about the importance of the molt and the cost of the molt in a variety of species. Use this information to set up some hypotheses. If there are clear hypotheses in the intro that may help engage the reader. Maybe 1st is to predict that they will not spend time in the water, and if they do, then you predict will spend more time in water at the end of the molt when the metabolic cost is lower (need to set this up) and more likely to be observed in lower condition seals. I think if the authors try to put this in a broader context both in the intro and discussion, the paper would be of interest to more readers.

My 2nd major concern is that the analyses are a bit superficial and more should be done to better interpret the results. As mentioned above, I think the paper would be greatly improved with hypotheses that you then test. As written, this paper is primarily descriptive and unfortunately often does not provide enough detail for the readers to correctly understand the data. I make many suggestions below on other potential analyses. This paper would be much stronger if you look at how body condition influences time spent in the water. I assume that you have access to this data as it is almost always collected when chemically immobilizing seals. At a minimum some sort of condition index with mass, length, and/or girth. If there is weather data that would also be important to include. The results presented are not necessarily analyzed or interpreted appropriately. When comparing mass loss between animals on land and in water they did not find a difference, yet they talk about it as if they did based on the figure. I did not find the figure very convincing and there are many variables that should be controlled for. Did the authors control for time spent in water? Weather? Stage of molt? I don't think it is valid to disagree with the stats – maybe just state that with limited size, and without controlling for important variables, you may be unable to detect a difference. It is also unclear why you spend so much time talking about the ingestion curves? If this is important you should set this up better in the intro. Comes off as a bit of a tangent. Overall, while I think it is great they detected feeding, this paper is written as a small note, not a full manuscript in a broad readership journal. In order to fit the scope of the journal the authors need to broaden the scope, do more analysis, and better place in current state of knowledge.

My last concern is the language. There are many confusing and vague sentences making it hard for the reader to get the main point. Sometimes ideas jump a bit too much so for a reader not familiar with the system, they would have a hard time following. I also found it difficult to understand the methods. I make more suggestions early on, but improvement in writing is needed throughout.

Specific concerns.

Abstract:

Line 23 – 26: The transition from ‘thought previously only on land for molt’ to ‘therefore we investigated’ is not strong. The fact that thought that on land does not seem like a strong justification for the study. Why is this important? Why might you suspect something different?

Line 29 – 30: Conclusion a bit weak – why don’t you evaluate the fitness consequences.

Intro:

Line 37: last phrase of the sentence is a bit vague/unclear, maybe expand just a bit. Important to have a powerful opening sentence.

Line 40: northern (and southern) should not be capitalized here and throughout

Line 41-43: Awkward sentence – language needs improvement – maybe – ‘‘Consequently, molting marine mammals exhibit strategies to minimize heat loss and the energetic cost of molting when on land, such as...’’ also habitat choices is a little vague – is not mud pools a type of habitat choice?

Line 43: Can make writing more concise and less passive, Ex. Many species fast and remain on shore for several weeks....

Line 45 – 48: Awkward sentence - try to rewrite so the point is clearer. Ex. Southern elephant seals are one of only 4 species of phocids that undergo a ‘‘catastrophic molt’’,(maybe in citation list the species with the citation?). As written you get lost in reading the 3 species that are not important in this paper – and the grammar is not quite correct.

Line 48 – 53: You can make this sentence more direct/concise – it is also unclear what you mean by constrained by high metabolic heat loss. Do you mean:

During the approximately 1-month molt, southern elephant seals maintain high peripheral vascular circulation to promote tissue growth, likely resulting in high metabolic costs due to heat loss (Add some citations – I know of at least one great Weddell seal one). Increased peripheral circulation is needed because the tissue grown requires a minimum temperature of 17 C and the mean ambient temperature in the sub-Antarctic island during the molting season (Jan-Feb) is ~8 C. This extensive heat loss to the cold environment leads to a molting metabolic rate that is estimated to be 2-3 times the resting metabolic rate. This heat loss would be even greater in the cold water where heat is lost xxx faster (I think you should add this last line to better set up your question and then this will not be no info in the next paragraph)

Line 54 – 59 – unclear and also a bit weird to have a one sentence paragraph. Maybe combine with following paragraph? You also mention large mass change for the first time here and you have not introduced that yet. People that do not know elephant seals may not know what you mean. Maybe develop this paragraph a bit more? Also - you don’t need to state without food or water intake or relying on body reserves– this is what fast mean so you are saying the same thing 3 times in one sentence. Try to be more concise to improve readability. This could be a nice spot to try to make more broad and talk about energetics and mass change in other species?

During the molt, elephant seals lose ~ XX% of their body mass. This large change in body mass, combined with the increased peripheral circulation resulting in high heat loss which would be exacerbated in water, led to the conclusion that elephant seals fast on land during the molt. However, few studies have examined the movement and potential feeding behaviour of molting southern

elephant seals.

Line 58-61: a bit of a jump. Try to connect better. You need to explain better since you will be using this technology. Just a sentence or 2 will help. . Also – at sea feeding has also been documented in at least one non-captive pinniped Kuhn, C. E., Crocker, D. E., Tremblay, Y., & Costa, D. P. (2009). Time to eat: measurements of feeding behaviour in a large marine predator, the northern elephant seal *Mirounga angustirostris*. *Journal of Animal Ecology*, 78(3), 513-523.

Line 62 – 67 – This paragraph is confusing and vague. This paragraph needs to be stronger to set up your study. Below is a quick suggestion, but it likely needs some work. I also was not familiar with the Boyd hypothesis and should not have to go look it up so I could not include that in my suggestion. You need to state the hypotheses – not just refer to it. I suggest setting up your study with specific hypotheses that you justify.

In this study we investigated the on-land and at-sea behavior of female southern elephant seals during the poorly studied molt. To do this we documented movement with xxx and ingestion events using stomach temperature.

Line 63: However is not the correct transition – maybe just delete.

Methods:

Line 72- 77: Try to be more concise with writing. Data were collected from 55 adult females during the austral summer molt (late Dec – late Feb) between 2014 and 2022. The fieldwork took place at 3 colonies within the Kerguelen Archipelago: Point Suzanne....

Line 87-88: You did not mention the location before. Do all tags do location? Just the splash. If you are providing location data, you need to explain how you got the location data and how you processed it – you provide maps in Supplemental, but you do not discuss this data.

Line 89-90: Much of this info was provided before – you don't need to repeat as much (don't need dates, or molting, etc). Maybe something more like: Of the 55 instrumented seals, we obtained stomach temperature data from 2+ days from 39 of the seals.

Line 95-96; You can't start a sentence with numerals – should be Eighteen of the instrumented seals were weighed at capture and recapture, allowing us to calculate daily mass loss (kg/day).

Line 101 – 125: I found this section confusing. The flow does not seem to make sense to me and I struggled to understand your definitions – even after referring to your figure. I see the black defines the dive, and the grey is the surface but you don't show what you considered a cycle. I don't think the first sentence adds anything. I am having a hard time suggesting something because I got pretty lost. I suggest reorganizing the Diving pattern section. Maybe start with how you processed the data (0 correction), then move on to how you defined a dive vs surface interval. And then once you defined the dive – what type of data did you extract (max depth, dive duration, time to max depth, etc). Once you clearly define a dive and post-dive interval, maybe then talk about the dive cycle? And highlight better on the figure. Because it was hard for me to figure out what you did – it makes it harder for me to evaluate the quality of the science (I am sure it is good – but you need to explain it better) From this point on – I will try to focus just on the validity of methods/content and not on writing – but writing still needs significant improvement.

Line 143-144: It is unclear to me why light level is important here? Maybe because I got a bit lost in the dive section? Please explain why seasonal decomposition of light-level matters. Also – data can not

be shorter. A study period can be. We could not analyze data from deployments shorter than 2 days....

Line 147 – you defined STP so why did you spell out again here? Also, I found this sentence a bit confusing - what do you mean by “as following...” And while you deemed drift was non-significant – you did not control for it. Suggest something more like: To examine STP temperature drift over time we down-sampled the data to one measurement every 15 minutes (factor of 100) and examined the trend in temperature over time using a Kendall’s tau correlation coefficient. The temperature increased for 16 individuals, decreased for 12 individuals, and did not change for the remaining 11. The mean.... Therefore, based on the ...

Line 162-170 I don’t think you need these 2 one sentence sections. Not ideal to have 1-2 sentence paragraphs and this information is already mentioned early in the methods.

Results

Line 198: “mean dives duration” should be “mean dive duration”

Line 202-203: A little confusing the way written – is the average duration for all females, or just the 47% that dived? Maybe: Seventy-seven percent of monitored females (n=30) performed surface swimming with an average duration of 1.8 +/- 2.6 hr, including 14 that also dived.

Line 204- 206: Awkward sentence, needs revising. I struggled a bit to understand – I make a suggestion but I am not 100% sure it is what you were trying to say. – maybe: Instrumented females on average spent time in the water (surface swimming or diving) every other day (% monitored days with at-sea events = 49.2 +/- 23.9%). The amount of time they spent in the water ranged from 9.0 – 92.8% of the day.

Line 208 – 210. Needs to be written more clearly. Also, if not significant you cannot say that they seem to be doing something different than what your results say. Your results indicate there is no difference. Maybe you can state with a larger sample size you could detect? I also don’t find the figure convincing of the statement. Yes the medians seem maybe a bit higher, but the boxes completely overlap.

Line 213-214. When you make statements like this there are usually stats to back it up. Maybe there are not enough events early on to do stats? Also, you do not provide enough information to interpret – can you control for time in the different phases. Maybe there are so few <40% molt events because you did not talk many seals is that phase.

Line 220 – when did the 34 other events happen? This is a bit too vague – on land? Need more information – why can you not say when they happened?

Line 221-227. Was that an average for all your seals? Or total for all 86 ingestions? Need to explain the result better.

Results section needs better reporting of results and a bit more analysis. The results currently reported are too simple, making this more like a note than a full manuscript. I am also not convinced of the importance because the analysis seems incomplete. There are other things you can hopefully report and analyses to do. I would like to understand more about your sample and at what phases did you instrument the seals. In one figure you state that there are more events later in the molt, which is not surprising to me, but is this because of when you instrumented the animals, or a change in behavior. I would like to know how many individuals were instrumented in <40% molt, 40-80, and

>90% molt. Can you see how condition influences likelihood of weather they feed? I think some studies in Wedds found that size or condition influences how likely to feed during lactation.

Discussion.

Line 243-244: This sentence is unclear and I don't think accurate. Not all marine mammals rely on body stores exclusively during molt. Fully aquatic animals and many pinnipeds (including phocid that don't have a catastrophic molt), continue to feed during the molt. You need to modify the system to reflect you are only referring to animals undergoing a catastrophic molt.

Line 259-261: You have a nice sample size of 55 – can you do more analysis with this data to try to understand why they went into the water? Were they sunny days? I have seen many northern elephant seals go into the water on sunny days – they are well insulated and with high solar radiation on calm days they seem to often move to wet kelp, wet sand, or even go into the shallows. Can you see if size or condition was a predictor of time spent in water?

Line 264-265 – Yes! Can you look at any of this? Do you have weather stations at the colonies? Are there weather records?

Line 266 – 267: If the result was not significant I don't really think this is a strong discussion point. The stats say it is not significant and the figure is not convincing. Also you have a lot of co-variables you are not exploring (maybe if you do you will find more support for this discussion). Can you look at time spent diving in relation to mass loss? Or the time in water in general? Did you control for when in the molt (early, mid late)? Weather can also influence metabolic rate? I suggest doing more analysis and if you don't find statistical support or at least more convincing results while controlling for some co-variables then this should be removed.

Line 280: Delete "do" before ingest, and "preys" should be "prey"

Line 287 – incorrect grammar – maybe: It is difficult to distinguish between water and prey ingestion events in pinnipeds.

Line 288 – can you come up with more informative name than Index I.

Line 297 – 306: Some good info in this paragraph, but many of the sentences are vague so hard to understand. It seems high that on average females ate 9.1 kg. Also, I am not sure how useful an average is when many animals did not have any or only a few ingestion events. I am finding it a bit hard to follow how you got to this number and what this really means. Is this dominated by just a few individuals that ate more?

Line 307 – 311. I think it is great you mention body condition, but I don't understand your logic – it may just be language. If in the previous study they thought it was water, why would it represent feeding? Maybe make it clear that based on water flux data the seals ingested x amount of water either through water or prey ingestion. If prey ingestion would = x amount of energy . Also - Do you have condition data on your seals? At a minimum some sort of index with mass/length? Or better condition estimates based on truncated cones methods? It is very unusual to chemically immobilize seals and not take basic morphometric measurements that can be used to estimate condition. If you have these data they should be included.

Line 312-318. This is great and likely true. There are papers with some of this information. I think your manuscript would be greatly improved if you do a bit more reading about molt in a variety of

pinnipeds. While different, there are some relevant studies on Weddell seals and a variety of pups. Currently, your paper is very elephant seal focus and I think you can make it more interesting to a broader range of readership by focusing on why this is important rather than just documenting they go in the water and feed.

Figures:

Figure 1: Need a better figure text. Is the whole time period in B, C and D a dive cycle – or are there multiple dive cycles? I am still unclear what a dive cycle is. A dive cycle is usually a dive and following surface interval, but that is not clear from your methods or figure text. Try to more clearly state what a dive is. Below is a suggestion based on how I would define a dive and dive cycle, but I am not sure if this is how you defined it. Also I don't think you need E – it is just a repeat of B. Would also be good to mention wet dry and light level and how you used them in this figure. Figures should be stand alone – i.e. the reader should not have to read the manuscript to understand your figure.

A dive is defined as a submergence below 5 m for a minimum duration of x seconds (black dashed line). A surface interval is the time between two dives (grey dashed line). A dive cycle is xxx (I usually would say dive and following surface interval – but not sure what you mean. If you mean a bout that is something different?)

Figure 2: would be better to use letters rather than numbers for subplots to be consistent with fig 1. Also best to present them in same order you mention them. (right now you have 3, then 1 and 2, then 4). Suggest moving #4 to #1 since you use this to explain a methods.

Table 1 would be more informative if you break up by molt state with sample size. I think we would see that almost all events were later in the molt stage? I feel like there is information missing from the paper that is needed to support your conclusions. See comments about the results earlier. How can an individual have 0.1 events? Did you calculate the mean or each seal? But not all seals had events?

Figure 3 – would be nice to include stats showing that there is no difference.

Figure 4 – I am confused by the figure text. You say each dot is an individual – but you have 40 black dots – I thought only 31 females had feeding events? Are some females represented twice because transitioned stages? Need to explain this. Do you have so few initial measurements because you did not instrument females early in the molt? Or did they just not go in the water early (by late in molt, there is less likely to be a big cost).

Figure 5 - figure text needs to be more clear. It is a bit of a run-on sentence. Also you use numbers but the subplots have letters. You need to better define what the unknown categories is? How do you not know when they are happening. If not in the ocean should you include them.

Some potentially relevant references

Walcott, S. M., Kirkham, A. L., & Burns, J. M. (2020). Thermoregulatory costs in molting Antarctic Weddell seals: impacts of physiological and environmental conditions. *Conservation physiology*, 8(1), coaa022.

Sato, K., Tsuchiya, Y., Kudoh, S., & Naito, Y. (2003). Meteorological factors affecting the number of Weddell seals hauling-out on the ice during the molting season at Syowa Station, East Antarctica.

Reviewer #2 (Remarks to the Author):

Review of the manuscript "Breaking the fast: First report of dives and ingestions events in molting Southern elephant seals".

This paper presents an original and extensive data set relying on the equipment of 55 southern elephant seals during the moult. These females were equipped with a time depth recorder, a stomach temperature pill and a satellite tag. Each female was monitored on average for 7.0 ± 2.9 days.

This species, is one of the three seal species experiencing a catastrophic moult (i.e. the renewal of both hair and part of the cornified epidermis). During the moult, elephant seals were considered as exclusively fasting extended period of fasting and in this species the moult period was previously thought to be spent entirely on land.

This study demonstrates that, contrarily to what was previously assumed, 77% of the elephant seal females went swimming at sea and 39% dove.

Strength of the paper

The results are original, change the perception on elephant seal ecology-ecophysiology, and may have strong implications on studies implying to estimate metabolic rate by implementing tritium or deuterium on that species during the moult.

The results will be definitely of interest for the marine biologist and marine mammal community as well as researchers studying fasting metabolisms.

The sample size is significant and therefore the main findings are robust.

Main weaknesses

Analyses could be developed regarding the influence of meteorological conditions, such as air temperature, wind, rain..., on the occurrence of such at-sea visit behaviour, were not investigated. I suppose such meteorological records must be available over the study period.

The diurnal pattern of at-sea visit was not investigated and this could provide valuable information about possible factors driving such behaviour. Is this behaviour only taking place exclusively during the day, at night or throughout the whole day?

I am not certain that mass in-itself is the most informative biometric factor. Depending on their sizes, seals with similar mass might be in fact in a very different state of body condition. Therefore, I would suggest the authors to perform the analyses on body condition changes rather than mass changes.

More importantly, the discussion of this paper remains very much elephant seal focussed. The general scope of this study needs to be broadened, regarding ecophysiological considerations such as peripheral perfusion and therefore fur synthesis. For instance, I got the impression that stomach temperature data are available both on land and at sea and, I would have found interesting to compare both stomach temperature, excluding stomach temperature drops, both on lands and at sea.

It is mentioned in the paper that Argos location were used to determine if the stomach temperature drops were taking place on land or at sea. Clearly according to the precision of the Argos location categories (3,2, 1, 0...) it is, most of the time, impossible to distinguish using this criteria if seal were really on land or at sea as the seals remain very close to shore. So I guess only the combination of Argos locations but in combination with the wet and dry sensor information, could help to assess the land versus at-sea period.

At the end it is unclear to me if some of these temperature drops were observed while the seals were on land, and if it is the case this is likely to suggest water ingestion. This could be used to compare the temperature drop signal from these land events versus at-sea events and possibly to be used to further discriminate between possible water versus prey ingestion events. If not such events are detected on land this should be clearly stated.

My general feeling is that much more could be done with this exceptional dataset, and at the end, the overall study could be more convincing.

Not being an English speaker myself, I cannot comment on the English grammar but I feel that the writing could be sharpened. I found many imprecisions throughout the document.

For these reasons, I recommend that this paper, if accepted, require to go through a major revision process.

Other comments:

Line 45 : angustirostris

Line 63: However, our data revealed at-sea movements and ingestion events that match with Boyd et al. hypothesis. This hypothesis should be clearly formulated here and not much later in the text. i.e. that prey and/water ingestion was likely to take place during the molt.

Line 123 :

Light-level variations due to the day-night cycle were removed using a seasonal decomposition function based on moving averages (function "seasonal decompose" from Python package "statsmodels" version 0.13.2.

This is unclear to me why this was done. Is this to distinguish between day and night?

Line 144 : Again it is unclear to me how and why the light level was used in this paper.

Line 156 : A visual check of stomach temperature curves over time for each ingestion led to the removal of anomalous events

How this was decided or defined as an anomalous event, this appears to be very suggestive.

Line 157 : "leading to an estimated rate of only 11% of false positives detected by our algorithm".

This should go in the result part. Above you mention a visual check and here an algorithm. This part remains very unclear to me. How much drinking took place why seals were on land.

Line 232, 234, 241: for the first time,. This study is the first, It is the first time...

A bit too much for me, once should be enough!

Line 246 to 248. This would have been better in the introduction than in the discussion.

Line 283-286 : Unclear to me how this was performed

The quality of the graph and figures require improvements.

Reviewer #3 (Remarks to the Author):

This manuscript represents a considerable and significant piece of work showing for the first time the evidence of breaking the fast during the energetically costly period of catastrophic molting for a marine mammal.

This work provides unique and critical cues on the ingestion events of Southern elephant seals and it was very interesting to see how the instruments were used to detect those diving and ingestion events. This research changes our perception of the molt for such predator and withdraws the hypothesis of complete fasting. This research is of immediate relevance for people in our field but also for a broader audience, as it discusses unexpected behavior and behavioral changes during a key period of the life cycle that could be interesting for both terrestrial and marine studies.

These observed ingestion events could provide valuable insights into the physiology of pinnipeds regarding their hydration during such costly period of their life cycle and their foraging behavior, very different from the rest of their diving behavior outside of the molt. It also represents an exceptional dataset for sub-antarctic pinniped species.

Analysis sounds robust, correct, and conducted in the right way. In the introduction, I would appreciate the authors extend on other marine seabird and mammals in terms of their molting strategy. The authors discussed it in the discussion section but more background in the introduction on other species would be greatly appreciated. In the methods section, the approach to differentiate water from prey ingestions should be clearly stated. In the results, I would appreciate adding a map of all tracking data covering the full period and coloring differently prey ingestion from water ingestion. The part that needs the most improvements is the discussion. I suggest providing 1) perspectives in order to improve the study to better quantify change in body mass, differences between water ingestion and prey ingestion; 2) a general discussion about potential preys found at such depths and their quality; 3) hypotheses regarding the consequences of such behavior on individual fitness, breeding success, and the role of inter-individual variability in such behavior; 4) the role of hydration for sub and Antarctic pinnipeds; 5) finally, I strongly suggest a short paragraph discussing the effect of temperature pills on the animal welfare. I understand it is not often easy to quantify the effect, but I recommend at least to provide some hypothesis on the potential effect of such pills.

I appreciated a very detailed description of the methods and the relevance of the figures.

Following those comments, I highly recommend this paper to be published with the suggested minor revisions. Please find below the minor edits requested on top of my general comments.

Abstract

- Unless this is a journal requirement, I would suggest the authors adding just a little quantitative information about the percentage of time spent feeding and ingesting water during the whole molting period.

Introduction

- L45-46: could the authors please put the latin names in parenthesis just after the species name.

Material and methods

- L86: could the authors please provide details about the retrieval of the temperature pills when not excreted?

- Figure S2: I suggest using a different color for the dots, for separating the 45% with estimated error radius <250 m from the other ones.

- L90: "9 of which were successfully associated with a STP for more than 2 days", could the authors please clarify, I am unsure of what the authors meant here with associated for more than 2 days?

- L95: the reference is not in the same format as other references.

- L138: "We then filtered out events that were too short in duration to be ingestions", how did the authors decide the threshold for duration?

- L144: please replace "those 6" by "those 6 individuals"

- L148: please replace "down sample" by "down-sampled"

- L185: it is not in Table S2 but figure S2?

-

Results

- It would be great to have a figure with all tracking data combined and colours would be the individuals

- Figure 4: why are they dots with the exact same value for at-sea events? Did the authors round the

results?

- L213: are these results significant? please specify whether it is or not. If not, the authors should be careful and say "the number of ... seems/tends to increase..."
- Figure 5: the legend mentions number of panels while the figure has letters, please correct this is confusing.
- L217-227: it is unclear how the authors differentiate from prey ingestion from water ingestion, I may have missed something, but this should be clarified and clearer.

Discussion

- L253: please add "and" in between "molt" and "also".
- L259-261: I suggest the authors to add some hypothesis regarding the prey they may target at such depths and their energetic quality.
- L266-271: given the data you have and the figure 3, I would not discuss any difference that are neither visible nor significant. I would comment more on the small sample size limiting the study and on the perspectives of what the authors should do to test this.
- L292-295: this information is very useful and should be clearly written in the methods section.
- L246: please change the wording "why".
- L308-310: please clarify, this is confusing, I did not get the message.
- L307-311: I suggest adding more words discussing the potential threshold of fat depletion that will lead to the decision to go at-sea. Moreover, it is not clear in the paragraph whether it is the need for water ingestion or feeding that motivate breaking the fast. I would also improve the discussion by adding a paragraph on the way the change in body fat could be monitored and improved, in order to identify those thresholds, very important to understand the behavior the authors are describing.
- L319-320: did the authors tested for this? I recommend testing it with surface temperatures on departure at-sea compared with the average at this period.
- L321: please remove "good"
- I suggest adding a paragraph at the end on hypothesis regarding the fitness consequences, breeding success...of such behavior (breaking the fast)

Sara Labrousse

Response to referees

Reviewer #1 (Remarks to the Author):

In this manuscript the authors report that some elephant seals spend time at sea, occasionally performing dives and feeding, during the molt. This is in contrast to the commonly held view that elephant seals remain on land and fast for the duration of the fast. While it is exciting that they have documented this with stomach temperature pills and data loggers, I do not think this manuscript is suitable for publication in *Communications Biology*. The authors need to provide more context on why this is important, should perform additional analysis, and improve the language so the reader can better understand the methods, results, and discussion. Due to vague language, not enough detail, and unclear writing it was difficult to understand all of the methods. Below I explain 3 major concerns followed by more detailed information about those concerns and language.

Major concerns:

My first major concern is that this paper is too elephant seal centric. The authors need to better set up the importance of molt in the intro. The paper would be much stronger if more background was provided about the importance of the molt and the cost of the molt in a variety of species. Use this information to set up some hypotheses. If there are clear hypotheses in the intro that may help engage the reader. Maybe 1st is to predict that they will not spend time in the water, and if they do, then you predict will spend more time in water at the end of the molt when the metabolic cost is lower (need to set this up) and more likely to be observed in lower condition seals. I think if the authors try to put this in a broader context both in the intro and discussion, the paper would be of interest to more readers.

My 2nd major concern is that the analyses are a bit superficial and more should be done to better interpret the results. As mentioned above, I think the paper would be greatly improved with hypotheses that you then test. As written, this paper is primarily descriptive and unfortunately often does not provide enough detail for the readers to correctly understand the data. I make many suggestions below on other potential analyses. This paper would be much stronger if you look at how body condition influences time spent in the water. I assume that you have access to this data as it is almost always collected when chemically immobilizing seals. At a minimum some sort of condition index with mass, length, and/or girth. If there is weather data that would also be important to include. The results presented are not necessarily analyzed or interpreted appropriately. When comparing mass loss between animals on land and in water they did not find a difference, yet they talk about it as if they did based on the figure. I did not find the figure very convincing and there are many variables that should be controlled for. Did the authors control for time spent in water? Weather? Stage of molt? I don't think it is valid to disagree with the stats – maybe just state that with limited size, and without controlling for important variables, you may be unable to detect a difference. It is also unclear why you spend so much time talking about the ingestion curves? If this is important you should set this up better in the intro. Comes off as a bit of a tangent. Overall, while I think it is great they detected feeding, this paper is written as a small note, not a full manuscript in a broad readership journal. In order to fit the scope of the journal the authors need to broaden the scope, do more analysis, and better place in current state of knowledge.

My last concern is the language. There are many confusing and vague sentences making it hard for the reader to get the main point. Sometimes ideas jump a bit too much so for a reader not familiar with the system, they would have a hard time following. I also found it difficult to

understand the methods. I make more suggestions early on, but improvement in writing is needed throughout.

Global answer

We thank the reviewer for highlighting the strengths and weaknesses of our manuscript, and for taking the time to make so many suggestions and rephrasing many parts of the paper. We followed the advices and added more context, more information about the molt and why it's an important event not only for seals. We also rewrote the introduction to introduce better the hypotheses and why we made them.

We also agree with the strength of the analyses. Please note that at first, the aim was to write a short report about at-sea travels and ingestions, before questioning the reasons for such behavior. After reading the comments, we agree it is more interesting to combine both in a single paper. Please find the new analyses described in the methods, results, and discussed. We included linear and generalized linear models to take the parameters suggested above (weather, stage of molt, monitoring length...). We also considered the BMI rather than the body mass as the size of the animal is important to consider. When we looked at body mass loss, we always added the length as a fixed variable in the model. We also rebalanced the results by spending less time to describe the ingestions but discussing the possible reasons for such behavior, as well as the physiological consequences.

Finally, we considered all the suggestions concerning the writing, and the final version has been checked by an English-native speaker.

Specific concerns.

Abstract:

1. Lines 23 -26: The transition from "thought previously only on land for molt" to "therefore we investigated" is not strong. The fact that thought that on land does not seem like a strong justification for the study. Why is this important? Why might you suspect something different? We thank the reviewer for highlighting this and changed the transition as followed (lines 22-24) "During the molt animals are therefore constrained by high metabolic heat loss and are thought to fast and remain on land.

To examine the ability of individuals to balance the energetic constraints of molting on land we investigated the stomach temperature and movement patterns of molting female SES."

2. Line 29 – 30: Conclusion a bit weak – why don't you evaluate the fitness consequences.

With the new analyses we evaluated the physiological consequences and discussed about fitness consequences in the discussion. We modified the abstract lines 28-29 as followed : "We conclude that the paradigm of fasting during the molt in this species, and the fitness consequences of this behavior should be reconsidered, especially in the context of a changing climate."

Intro:

3. Line 37: last phrase of the sentence is a bit vague/unclear, maybe expand just a bit. Important to have a powerful opening sentence.

We changed the introduction specifying the importance of the molt in the animal kingdom, lines 41-51, as followed:

"Molting is an important physiological event of partial or total renewal of integuments and influenced by season, growth, age, and environmental conditions. This process occurs in a large variety of species, including mammals, birds, reptiles, fishes and arthropods. Molting is important for an individual's fitness a change in coat or plumage color may be related to reproductive

success and camouflage in the environment, growth, thermoregulation, and defense against parasites or pathogens. However, molt is an energy demanding process requiring high peripheral vascular circulation to promote skin cell proliferation and tissue growth, and requiring high metabolic rate to balance heat loss. Many phocid seals haul out of water for extended periods during their annual molt, when they shed and regrow their pelage. This behavior is believed to limit heat loss to the environment given increased peripheral blood flow to support tissue regeneration. The degree to which time in water, particularly during the molt, may affect thermoregulatory costs is poorly understood."

4. Line 40: northern (and southern) should not be capitalized here and throughout.
We corrected the writing line 63 and throughout in the manuscript.

5. Lines 41-43: Awkward sentence - language needs improvement - maybe - "Consequently, molting marine mammals exhibit strategies to minimize heat loss and the energetic cost of molting when on land, such as..." also habitat choices is a little vague - is not mud pools a type of habitat choice?

We added details about seals adaptations as followed (lines 56-60):

"Maintaining a high surface temperature in a cold environment results in high metabolic costs; with molting metabolic rate estimated to increase by a factor of 2-3 compared to resting metabolic rate. To deal with this high energy expenditure, phocid seals are well insulated with a thick subcutaneous blubber; regulate peripheral blood flow; and avoid high heat loss in cold water by hauling out onto land more frequently during the molt."

We also added habitat types lines 72-73: "SES therefore minimize heat loss through social thermoregulation in large aggregations and through choice of different beach habitats such as grass or mud pools."

6. Line 43: Can make writing more concise and less passive, Ex. Many species fast and remain on shore for several weeks...

We rewritted more concise as suggested lines 55-60: "To deal with this high energy expenditure, phocid seals are well insulated with a thick subcutaneous blubber; regulate peripheral blood flow; and avoid high heat loss in cold water by hauling out onto land more frequently during the molt."

7. Lines 45 - 48: Awkward sentence - try to rewrite so the point is clearer. Ex. Southern elephant seals are one of only 4 species of phocids that undergo a "catastrophic molt",(maybe in citation list the species with the citation?). As written you get lost in reading the 3 species that are not important in this paper - and the grammar is not quite correct.

Sentence was rewritten as suggested with line 61-64: "Southern elephant seals (SES) are one of only four species of phocids that undergo a 'catastrophic molt', the only one in a polar environment, during which they renew both their hair and cornified epidermis (with northern elephant seal (NES), *Mirounga angustirostris*, Hawaiian monk seal (*Monachus schauinslandi*), and Mediterranean monk seal (*Monachus monachus*))."

8. Line 48 - 53: You can make this sentence more direct/concise - it is also unclear what you mean by constrained by high metabolic heat loss. Do you mean:

During the approximately 1-month molt, southern elephant seals maintain high peripheral vascular circulation to promote tissue growth, likely resulting in high metabolic costs due to heat loss (Add some citations - I know of at least one great Weddell seal one). Increased peripheral circulation is needed because the tissue grown requires a minimum temperature of 17 C and the mean ambient temperature in the sub-Antarctic island during the molting season (Jan-Feb) is ~8 C. This extensive heat loss to the cold environment leads to a molting metabolic rate that is

estimated to be 2-3 times the resting metabolic rate. This heat loss would be even greater in the cold water where heat is lost xxx faster (I think you should add this last line to better set up your question and then this will not be no info in the next paragraph)

We thank the reviewer for the suggested reference. We added "heat is lost 25 time faster" and rewrite the paragraph as follow: "" (lines 52-60): "Most Antarctic and sub-Antarctic phocid seals undergo an annual molt. While seals are well adapted for an aquatic life, they are obliged to come ashore (on land or ice floe) to molt. In phocids, the maintenance of a warm skin for extended periods for the shedding and renewal of hair is required as tissue growth ceases below 17°C, which is inconsistent with permanent cold water exposure where heat is lost 25 times faster than in air. Maintaining a high surface temperature in a cold environment results in high metabolic costs; with molting metabolic rate estimated to increase by a factor of 2-3 compared to resting metabolic rate. To deal with this high energy expenditure, phocid seals are well insulated with a thick subcutaneous blubber; regulate peripheral blood flow; and avoid high heat loss in cold water by hauling out onto land more frequently during the molt."

9. Lines 54 - 59 - unclear and also a bit weird to have a one sentence paragraph. Maybe combine with following paragraph? You also mention large mass change for the first time here and you have not introduced that yet. People that do not know elephant seals may not know what you mean. Maybe develop this paragraph a bit more? Also - you don't need to state without food or water intake or relying on body reserves– this is what fast mean so you are saying the same thing 3 times in one sentence. Try to be more concise to improve readability. This could be a nice spot to try to make more broad and talk about energetics and mass change in other species? During the molt, elephant seals lose ~ XX% of their body mass. This large change in body mass, combined with the increased peripheral circulation resulting in high heat loss which would be exacerbated in water, led to the conclusion that elephant seals fast on land during the molt. However, few studies have examined the movement and potential feeding behaviour of molting southern elephant seals.

We thank the reviewer for pointing out the need for more information about mass changes in other pinnipeds. We considered the rephrasing suggestion and developed the paragraph lines 66-67 by adding mass changes reported in northern elephant seals, southern elephant seals, grey seals: ". SES and other phocids lose a large proportion of their body mass during the molt (23% SES, 25%, NESs, 17%, gray seals *Halichoerus grypus*). This large change, combined with the increased peripheral circulation which would result in high heat loss in water, led to the conclusion that elephant seals fast and remain on land during the molt."

10. Line 58-61: a bit of a jump. Try to connect better. You need to explain better since you will be using this technology. Just a sentence or 2 will help. . Also – at sea feeding has also been documented in at least one non-captive pinniped Kuhn, C. E., Crocker, D. E., Tremblay, Y., & Costa, D. P. (2009). Time to eat: measurements of feeding behaviour in a large marine predator, the northern elephant seal *Mirounga angustirostris*. *Journal of Animal Ecology*, 78(3), 513-523.

We added more information about the method lines 81-87 : "One way to investigate at-sea feeding or dinking behavior is to combine stomach temperature measurements with time-depth recorders (TDRs). This allows detection of possible cold water or prey item ingestion through stomach temperature changes and to record the depth and location of ingestion events. This approach has previously been used in seabird research (penguins, cormorants, albatrosses⁴¹⁻⁴⁴, and captive or non-captive pinnipeds (NESs and Californian sea lions^{24,45}), but not during the molt. Physiological and meteorological factors are known to impact heat loss and molting phenology in phocids^{15,46}, and therefore both factors are required to fully understand the molt in SES."

11. Lines 62 - 67 - This paragraph is confusing and vague. This paragraph needs to be stronger to set up your study. Below is a quick suggestion, but it likely needs some work. I also was not familiar with the Boyd hypothesis and should not have to go look it up so I could not include that in my suggestion. You need to state the hypotheses – not just refer to it. I suggest setting up your study with specific hypotheses that you justify.

In this study we investigated the on-land and at-sea behavior of female southern elephant seals during the poorly studied molt. To do this we documented movement with xxx and ingestion events using stomach temperature.

We rephrased the sentence as suggested lines “In this study, we therefore investigated on-land and at-sea behavior of female SES while molting. To do this, we documented diving behavior with TDRs and ingestion events using stomach temperature. We expected that female SES would remain on land, in mud pools, or on the beach, fasting, at least in the early stages of the molt to minimize heat loss. Also, we wanted to find out if females went to sea to drink or feed during the molt, and if weather conditions influenced this behavior. If animals went to sea then we predicted that only females in good body condition would have sufficient energy reserves to compensate for high heat loss from the skin when molting.”

We also introduced the Boyd hypothesis line 78-81: “Although water ingestion has been suggested as a mechanism for SES to mitigate physiological costs while molting, there have been few studies examining movement and potential drinking or feeding behavior in this species throughout the molt.”

12. Line 63: However is not the correct transition – maybe just delete. We deleted as suggested.

Methods:

13. Line 72- 77: Try to be more concise with writing. Data were collected from 55 adult females during the austral summer molt (late Dec – late Feb) between 2014 and 2022. The fieldwork took place at 3 colonies within the Kerguelen Archipelago: Point Suzanne....

We rephrased as suggested line 99-103 so this part is more concise: “All scientific procedures were approved by the Ethics Committee of the French Polar Institute. Data were collected during the austral summer molt (December-February) between 2014 and 2022. The field work took place at three colonies within the Kerguelen Archipelago: Pointe Suzanne (49°26’S, 70°26’E) between 2014 and 2019, Estacade (49°16’S, 70°32’E) in 2020 and 2022, and Port-Aux-Français (49°34’S, 70°21’E) in 2021.”

14. Line 87-88: You did not mention the location before. Do all tags do location? Just the splash. If you are providing location data, you need to explain how you got the location data and how you processed it – you provide maps in Supplemental, but you do not discuss this data.

We specified which tag does location. We kept Figure S1 to show a good quality at-sea travel. However, because of few and poor quality Argos locations, we did not use them to analyze diving behavior and ingestion events. We combined depth, wet/dry and light level data to do so. As it was confusing, we clarified (see below) and deleted Fig S2 as it brought nothing more than S1. Method section (lines 104-108): “Individuals were equipped with devices provided by Wildlife Computers (USA): Time-Depth Recorders (TDR10-LX-340, 57 x 38 x 30 mm, 75 g in air), Argos satellite transmitters (SPLASH10-L-309D, 76 x 55 x 32 mm, 125 g in air) and Stomach Temperature Pills (TDR-STP-207D, 63 x 21 mm, 31 g in air). Argos location was obtained each day from resulting transmissions. Locations are classified as 3, 2, 1, 0 according to the error radius (<250m, 250-500m, 500-1500m, > 1500m). Due to short monitoring and poor Argos location quality, we did not use these data in diving pattern analysis and ingestion events. We show an example of good at-sea locations in Figure S1.”

15. Line 89-90: Much of this info was provided before – you don’t need to repeat as much (don’t

need dates, or molting, etc). Maybe something more like: Of the 55 instrumented seals, we obtained stomach temperature data from 2+ days from 39 of the seals.

We thank the reviewer for the suggestion and rephrased as advised lines 119-121.

16. Line 95-96; You can't start a sentence with numerals – should be Eighteen of the instrumented seals were weighed at capture and recapture, allowing us to calculate daily mass loss (kg/day).

We rewrote as suggested line 123-126: "Eighteen of the instrumented seals were weighed at capture and recapture, allowing us to calculate the daily mass loss (kg/day). Length (m) was calculated from the mean length measured at capture and recapture. From this, we calculated a body mass index (BMI, kg/m²)."

17. Line 101 – 125: I found this section confusing. The flow does not seem to make sense to me and I struggled to understand your definitions – even after referring to your figure. I see the black defines the dive, and the grey is the surface but you don't show what you considered a cycle. I don't think the first sentence adds anything. I am having a hard time suggesting something because I got pretty lost. I suggest reorganizing the Diving pattern section. Maybe start with how you processed the data (0 correction), then move on to how you defined a dive vs surface interval. And then once you defined the dive – what type of data did you extract (max depth, dive duration, time to max depth, etc). Once you clearly define a dive and post-dive interval, maybe then talk about the dive cycle? And highlight better on the figure. Because it was hard for me to figure out what you did – it makes it harder for me to evaluate the quality of the science (I am sure it is good – but you need to explain it better)

From this point on – I will try to focus just on the validity of methods/content and not on writing – but writing still needs significant improvement.

We thank the reviewer for pointing out the confused organization of this paragraph. We rewrote as suggested lines 131-147 and hope it is now easier to understand. We improved writing and asked an English native speaker to check.

"Because Argos locations were poor, diving patterns were analyzed only from depth, light level, and wet/dry sensor data. When necessary, a depth-zero correction was made on land at the time of capture. Diving behavior was determined using a Python algorithm especially developed for this study (Hex-Data, France). To make sure light level variations involved depth and not daily changes, day-night light-level variations were removed using a seasonal decomposition function based on moving averages (function "seasonal decompose" from Python package "statsmodels"; version 0.13.2). We combined a depth threshold of 0.5 m and a "wet" display to spot the start of a dive. To distinguish mud pool (wallows) from sea-water baths, we made sure the light level varied by more than 5 units (values ranging from 0 to 255) during the presumed dive.

At sea behavior was characterized by 1) a submerged phase, 2) a maximum depth, and 3) a surface phase. We distinguished surface swimming and diving behavior according to maximum depth. Surface swimming was defined to occur between 1 and 5 m, with a maximum surface time of 20 minutes between two submerged phases (Figure 1A). Dives were described by a descent, a bottom, and an ascent phase with a maximum depth greater than 5 m and showed different profiles (Figures 1B, 1C, 1D). A dive cycle was characterized by at least two successive dives separated by less than 20 minutes at the surface, as surface duration intervals showed a break around 17 minutes. Events were described by their mean duration, mean and maximum depth, and frequency."

18. Line 143-144: It is unclear to me why light level is important here? Maybe because I got a bit lost in the dive section? Please explain why seasonal decomposition of light-level matters. Also – data can not be shorter. A study period can be. We could not analyze data from deployments shorter than 2 days....

We thank the reviewer for noticing light level use was confusing. To make sure light level variations were associated with dives, we filtered the data to remove the day-night variations. We rewrote the section lines 134-139: "). To make sure light level variations involved depth and not daily changes, day-night light-level variations were removed using a seasonal decomposition function based on moving averages (function "seasonal decompose" from Python package "statsmodels"; version 0.13.2)."

We corrected the sentence and changes data by period as suggested line 171: "Six animals with less than 2 days of recording were discarded".

19. Line 147 – you defined STP so why did you spell out again here? Also, I found this sentence a bit confusing - what do you mean by "as following..." And while you deemed drift was non-significant – you did not control for it. Suggest something more like: To examine STP temperature drift over time we down-sampled the data to one measurement every 15 minutes (factor of 100) and examined the trend in temperature over time using a Kendall's tau correlation coefficient. The temperature increased for 16 individuals, decreased for 12 individuals, and did not change for the remaining 11. The mean.... Therefore, based on the ...

We thank the reviewer for pointing this and checked to use STP throughout the text.

We also rewrote the section as suggested lines 163-170: "To examine STP temperature drift over time, we downsampled the data to one measurement every 15 minutes (factor of 100) and examined the trend in temperature over time using Kendall's tau correlation coefficient. The temperature increased for 16 individuals, decreased for 12 individuals, and did not change for the remaining 11 individuals. The mean absolute value of the correlation coefficient was $\tau = 0.11 \pm 0.09$, with the mean absolute value of the slope estimator $s = 5.1 \cdot 10^{-7} \pm 6.2 \cdot 10^{-7}$, corresponding to a variation of $\pm 0.1^\circ\text{C}$ every 10 days (for 72% of the 39 individuals; $n = 31$). Therefore, based on the temperature pill accuracy ($\pm 0.3^\circ\text{C}$) and mean survey duration (over a week), we considered that the signal drift was trivial and chose not to apply a correction."

21. Line 162-170 I don't think you need these 2 one sentence sections. Not ideal to have 1-2 sentence paragraphs and this information is already mentioned early in the methods.

We deleted this section and reported the information early in the methods (lines 123-126: "Eighteen of the instrumented seals were weighed at capture and recapture, allowing us to calculate the daily mass loss (kg/day). Length (m) was calculated from the mean length measured at capture and recapture. From this, we calculated a body mass index (BMI, kg/m^2)")

Results

22. Line 198: "mean dives duration" should be "mean dive duration"

We corrected line 241.

23. Line 202-203: A little confusing the way written – is the average duration for all females, or just the 47% that dived? Maybe: Seventy-seven percent of monitored females ($n=30$) performed surface swimming with an average duration of 1.8 ± 2.6 hr, including 14 that also dived.

We modified the sentence as suggested line 245: "We detected a total of 375 surface swimming events with an average duration of 1.8 ± 2.6 hours grouped in 226 surface swimming cycles. Seventy-seven percent of monitored females ($n = 30$) performed surface swimming, including 15 that also dived."

24. Line 204- 206: Awkward sentence, needs revising. I struggled a bit to understand – I make a suggestion but I am not 100% sure it is what you were trying to say. – maybe: Instrumented females on average spent time in the water (surface swimming or diving) every other day (%

monitored days with at-sea events = $49.2 \pm 23.9\%$). The amount of time they spent in the water ranged from 9.0 – 92.8% of the day.

We thank the reviewer for taking the time to rephrase the part that was unclear. We modified the sentence as suggested line 248-250.

25. Line 208 – 210. Needs to be written more clearly. Also, if not significant you cannot say that they seem to be doing something different than what your results say. Your results indicate there is no difference. Maybe you can state with a larger sample size you could detect? I also don't find the figure convincing of the statement. Yes the medians seem maybe a bit higher, but the boxes completely overlap.

We thank the reviewer for pointing this and suggesting to make other analyses. The effect of at-sea travel and ingestions on body mass loss was investigated using mixed linear models to consider monitoring duration, molting stage at equipment, and length of the females. We detailed these analyses in the method section (lines 216-227) and results (line 273-283).

26. Line 213-214. When you make statements like this there are usually stats to back it up. Maybe there are not enough events early on to do stats? Also, you do not provide enough information to interpret – can you control for time in the different phases. Maybe there are so few <40% molt events because you did not talk many seals is that phase.

We thank the reviewer for the suggestions and reported the females' molting stage at the beginning of the monitoring in the result section (lines 274-275): "Molt stage when animals were equipped was $54 \pm 25\%$ with 41% of the females caught at the initial stage ($n = 16$), 41% at the mid stage ($n = 16$), and 18% at the final stage of molt ($n = 7$). Body mass at the end of the monitoring averaged 294.5 ± 37.7 kg, resulting in a daily body mass loss of 3.2 ± 1.4 kg per day. At this point all females were at the final stage of molt, with more than 90% of hair renewed." We also investigated the effect of the molting stage at equipment on our female's behavior through mixed linear models. We detailed the analyses in the methods (lines 216-227) and results (line 273-283).

27. Line 220 – when did the 34 other events happen? This is a bit too vague – on land? Need more information – why can you not say when they happened?

We clarified line 266-268. These events appeared to happen on land but due to unprecise Argos location, and because animals used to remain close to the sea, we struggle knowing if they were on land or at sea. We looked at the wet sensor and light level, but they did not allow us to say animals were at sea. Based on field observations, our hypothesis is that animals could have ingest water but without being submerged (only the mouth inside the water). As they spend a lot of time in the tidal zone, they could ingest water without being submerged.

28. Line 221-227. Was that an average for all your seals? Or total for all 86 ingestions? Need to explain the result better.

We thank the reviewer for pointing this and clarified lines 262-270: "A summary of ingestion event variables is available in Table S1. We recorded a total of 87 ingestion events (Table 2), from 80% of individuals ($n = 31$ from total 39), with an average of 2.8 ± 2.1 ingestions per individual. Among these events, 70% occurred during the day.

A total of 10 ingestion events were detected during a dive (12%) and 42 events occurred during surface swimming (49%). Due to bad Argos locations, the remaining 34 events (39%) were not clearly characterized as at sea or on land, but were not associated with complete submersion of the animal according to wet and light sensors. These ingestion events could have happened strictly on land or in the tidal zone.

Results of the different methods based on the literature⁴⁵ are reported in Table 2. Based on the different methods, 24 to 51% of the ingestions were classed as potential prey ingestions.”

29. Results section needs better reporting of results and a bit more analysis. The results currently reported are too simple, making this more like a note than a full manuscript. I am also not convinced of the importance because the analysis seems incomplete. There are other things you can hopefully report and analyses to do. I would like to understand more about your sample and at what phases did you instrument the seals. In one figure you state that there are more events later in the molt, which is not surprising to me, but is this because of when you instrumented the animals, or a change in behavior. I would like to know how many individuals were instrumented in <40% molt, 40-80, and >90% molt. Can you see how condition influences likelihood of weather they feed? I think some studies in Wedds found that size or condition influences how likely to feed during lactation.

We thank the reviewer for the suggestion. As specified in comment 26., we added the information on the molting stage at equipment in the result section (lines 274-275). Also, we investigated the effect of body condition (with Body Mass Index calculation) on female’s behavior (going at-sea, ingestions). Analyses are described in the methods (lines 216-227) and results (line 273-283), Figure 4.

Figure 4: Daily mass specific delta mass (g/kg/day) of female southern elephant seals according to : A. Their diving score (PC1 as individual values on the first component of the PCA) and B. The number of ingestions. Individuals with high PC1 values perform more dives, that last longer, and more surface swimming.

Discussion

30. Line 243-244: This sentence is unclear and I don't think accurate. Not all marine mammals rely on body stores exclusively during molt. Fully aquatic animals and many pinnipeds (including phocid that don't have a catastrophic molt), continue to feed during the molt. You need to modify the system to reflect you are only referring to animals undergoing a catastrophic molt.

We thank the reviewer for pointing this and totally agree; we modified the sentence as follow:

"In marine mammals, pinnipeds undergoing such an extreme physiological event, including elephant seals, have always been thought to rely exclusively on energy reserves in blubber during the molt." (lines 312-314)

31. Line 259-261: You have a nice sample size of 55 – can you do more analysis with this data to try to understand why they went into the water? Were they sunny days? I have seen many northern elephant seals go into the water on sunny days – they are well insulated and with high solar radiation on calm days they seem to often move to wet kelp, wet sand, or even go into the shallows. Can you see if size or condition was a predictor of time spent in water?

We thank the reviewer for the suggestions. We included meteorological data in the analyses, and used mixed generalized linear models to investigate their effect on the proportion of females going at sea. We investigated the effect of insulation duration (min), windspeed (m/s), relative humidity (%) and ambient temperature (°C) on surface swimming behavior, diving behavior, and ingestion events. Methods are described lines 229-233, results lines 285-295, and in Table 5, Figure 5.

32. Line 264-265 – Yes! Can you look at any of this? Do you have weather stations at the colonies? Are there weather records?

We investigated the effect of the weather (see above, answer to comment 31). Methods are described lines 229-233, results lines 285-295, and in Table 5, Figure 5.

33. Line 266 – 267: If the result was not significant I don't really think this is a strong discussion point. The stats say it is not significant and the figure is not convincing. Also you have a lot of co-variates you are not exploring (maybe if you do you will find more support for this discussion). Can you look at time spent diving in relation to mass loss? Or the time in water in general? Did you control for when in the molt (early, mid late?)? Weather can also influence metabolic rate? I suggest doing more analysis and if you don't find statistical support or at least more convincing results while controlling for some co-variates then this should be removed.

We thank the reviewer for the suggestion. We deleted this section and added new analyses. We considered monitoring duration and molting stage at equipment in the models to investigate the time in water. We described them in the method section lines 203-227.

34. Line 280: Delete "do" before ingest, and "preys" should be "prey"

We corrected and rephrased lines 298-299 : "Through the use of time-depth recorders and stomach temperature loggers, we were able to identify surface swimming, diving, and ingestion events during the molt of female SES."

35. Line 287 – incorrect grammar – maybe: It is difficult to distinguish between water and prey ingestion events in pinnipeds.

We thank the reviewer for rephrasing, and corrected as suggested line 335.

36. Line 288 – can you come up with more informative name than Index I.

We added the significance of Index I and called the method sections for more details as followed (lines 335-336): "In their experimental study, index I (index of the rate of stomach temperature recovery, see Method section) values for prey ingestion ranged from 55.1 to 4380.0 s.°C⁻¹, while water ranged from 37.8 to 764.0 s.°C⁻¹."

37. Line 297 – 306: Some good info in this paragraph, but many of the sentences are vague so hard to understand. It seems high that on average females ate 9.1 kg. Also, I am not sure how useful an average is when many animals did not have any or only a few ingestion events. I am

finding it a bit hard to follow how you got to this number and what this really means. Is this dominated by just a few individuals that ate more?

Due to the difficulty to determine water or prey ingestion, and due to shallow dives, we decided to go through the main results and only include the type of ingestion, without considering the size. As we added many other results (meteorological data, body condition, state of molt), we wanted to focus on the main results to keep the reader focused. We rewrote the section lines 335-345).

38. Line 307 – 311. I think it is great you mention body condition, but I don't understand your logic – it may just be language. If in the previous study they thought it was water, why would it represent feeding? Maybe make it clear that based on water flux data the seals ingested x amount of water either through water or prey ingestion. If prey ingestion would = x amount of energy . Also - Do you have condition data on your seals? At a minimum some sort of index with mass/length? Or better condition estimates based on truncated cones methods? It is very unusual to chemically immobilize seals and not take basic morphometric measurements that can be used to estimate condition. If you have these data they should be included.

We added BMI index (calculated as body mass / size²) to get a better proxy for body condition. When we investigated body mass changes, we included the length of the animal as a fixed factor in the models. We detailed the analyses in the method section (lines 216-227) and results (line 273-283).

39. Line 312-318. This is great and likely true. There are papers with some of this information. I think your manuscript would be greatly improved if you do a bit more reading about molt in a variety of pinnipeds. While different, there are some relevant studies on Weddell seals and a variety of pups. Currently, your paper is very elephant seal focus and I think you can make it more interesting to a broader range of readership by focusing on why this is important rather than just documenting they go in the water and feed.

We thank the reviewer for the suggested papers. We broadened the scope to other molting pinnipeds, to discuss meteorological and physiological implications in southern elephant seal's behavior. We rewrote the discussion section in a broader range to focus on molting rather than elephant seals (second paragraph, line 351-407 and third paragraph, line 411-441).

Figures:

40. Figure 1: Need a better figure text. Is the whole time period in B, C and D a dive cycle – or are there multiple dive cycles? I am still unclear what a dive cycle is. A dive cycle is usually a dive and following surface interval, but that is not clear from your methods or figure text. Try to more clearly state what a dive is. Below is a suggestion based on how I would define a dive and dive cycle, but I am not sure if this is how you defined it. Also I don't think you need E – it is just a repeat of B. Would also be good to mention wet dry and light level and how you used them in this figure. Figures should be stand alone – i.e. the reader should not have to read the manuscript to understand your figure.

A dive is defined as a submergence below 5 m for a minimum duration of x seconds (black dashed line). A surface interval is the time between two dives (grey dashed line). A dive cycle is xxx (I usually would say dive and following surface interval – but not sure what you mean. If you mean a bout that is something different?)

We thank the reviewer for the comment and for taking the time to rewrite a definition of a dive. We implemented the method section with more precise definitions (lines 140-147). We also modified the figure by deleting the bottom part which was just a biggest B. We changed the text to make it clearer as followed :

Figure 1: Diving parameters during different at-sea behaviors; diving or surface swimming. Surface swimming occurs between 1 and 5 m, while a dive is characterized by a maximal depth of more than 5 m. A dive or a surface swimming cycle is characterized by at least two successive dives or surface swimming separated by less than 20 min at the surface. Fig 1A. represents a surface swimming cycle and different profiles of dive cycles (B, C, D) of a female southern elephant seal in 2017. Each dive is defined by the succession of a submerged phase (black dotted two-way arrow) and a surface phase (red two-way arrow) (B).

41. Figure 2: would be better to use letters rather than numbers for subplots to be consistent with fig 1. Also best to present them in same order you mention them. (right now you have 3, than 1 and 2, then 4). Suggest moving #4 to #1 since you use this to explain a methods. We thank the reviewer for the suggestion and modified the Figure as advised (see below).

Figure 2: Stomach temperature variations ($^{\circ}\text{C}$) during ingestion events (standard (A), double (B, C) or extreme (D)) recorded for two female Southern elephant seals in 2014 and 2019. T_1 is the initial temperature, T_2 the minimal temperature, T_3 the recovery temperature, ΔT the delta of temperature between T_1 and T_2 , $T_{0.5}$ the half-way recovery temperature between T_1 and T_2 ($T_{0.5} = T_2 + \Delta T/2$) and $t_{0.5}$

the time of half-way recovery (black arrow: $t_{0.5} = t_{T_{0.5}} - t_{T_1}$). The area above the curve (integral) was calculated from the asymptote.

42. Table 1 would be more informative if you break up by molt state with sample size. I think we would see that almost all events were later in the molt stage? I feel like there is information missing from the paper that is needed to support your conclusions. See comments about the results earlier. How can an individual have 0.1 events? Did you calculate the mean or each seal? But not all seals had events?

We thank the reviewer for pointing this. We added in the results section the molting stage of the individuals at the equipment as follow "Molting stage at equipment was 54 ± 25 % with 41% of the females equipped at initial stage (n=16), 41% at mid stage (n=16), and 18% at final stage (n=7) [...] All the females were at the final stage with more than 90% of the hair renewed at the end of the monitoring." However, as a daily visual check was not always possible, regular information on their molting stage throughout the monitoring was hard to get. Also, as molting advancement is not linear (especially if seals go at sea, which can slower the molt), it was difficult to get reliable information based on mathematical estimations. However, to consider the effect of the molting stage in the analyses, we added the molting stage at equipment and the monitoring length (because all the animals were at the final stage when tags were retrieved).

43. Figure 3 – would be nice to include stats showing that there is no difference.

We deleted Figure 3 as we completely changed the analyses. Figure 3 is now a PCA plot to show the groups according to at-sea behavior. Figure 4 represents the results of linear models

described in the method section 'lines 216-227) and Table 3. As their effect was significant in our models, we found more interesting to plot the daily mass specific delta mass (g/kg/day) according to : A. Their diving score (PC1 as individual values on the first component of the PCA) and B. The number of ingestions.

Table 3. Anova table of the best linear mixed effect models for at-sea behavior (PC1 and PC2) and physiological consequences (body mass loss).

Group	Parameter	AICc	Explanatory variables	X ²	Df	P
Diving	PC1	63.3	Monitoring length	13.43	1	< 0.001 ***
Surface swimming	PC2	56.5	Monitoring length	0.34	1	0.56
Diving + Surface swimming	Body mass loss	126.7	Mass start	0.02	1	0.89
			Length	0.37	1	0.54
			PC1	8.19	1	0.004 **
			Number of ingestions	4.42	1	0.03 *
			Monitoring length	10.65	1	0.001 **
On land	Body mass loss		Mass start	0.38	1	0.53
			Length	0.27	1	0.61
			Monitoring length	1.41	1	0.24

Figure 4: Daily mass specific delta mass (g/kg/day) of female southern elephant seals according to : A. Their diving score (PC1 as individual values on the first component of the PCA) and B. The number of ingestions. Individuals with high PC1 values perform more dives, that last longer, and more surface swimming.

44. Figure 4 – I am confused by the figure text. You say each dot is an individual – but you have 40 black dots – I thought only 31 females had feeding events? Are some females represented twice because transitioned stages? Need to explain this. Do you have so few initial measurements because you did not instrument females early in the molt? Or did they just not go in the water early (by late in molt, there is less likely to be a big cost).

We thank the reviewer for questioning the relevance of the figure. We deleted this one and reported the molting stage at equipment in the result section (lines 273-277). As explained above, getting the molting stage associated with each at-sea event or ingestion event was not possible, but we included the molting stage at equipment in our models as well as the monitoring length to consider their effect on our females' behavior.

45. Figure 5 - figure text needs to be more clear. It is a bit of a run-on sentence. Also you use numbers but the subplots have letters. You need to better define what the unknown categories is? How do you not know when they are happening. If not in the ocean should you include them.

We thank the reviewer for pointing out this. The unknow categories refers to ingestions that were not at-sea (animal not under water according to wet and light sensor), but for which Argos location remain uncertain. Animals could be on land or in the tidal zone, drinking water without being under the sea. As the aim is to compare the different methods used to detect feeding events, we choose to delete this figure and to compare the 4 methods in Table 2 (see below), without distinguish where they occurred.

Table 2: Summary of ingestion type based on Kuhn and Costa (2006) results, from 31 female southern elephant seals monitored between 2014 and 2022 (n = 87 events).

Method	Threshold for feeding events	Number of feeding events
Index of rate of temperature recovery I (s.°C ⁻¹)	I value > 250 s.°C ⁻¹	n = 37 (43%)
Area above the curve (s.°C)	Integral > 3000 s.°C	n = 24 (28%)
Time of temperature recovery (Δt between B-C)	$t_{rec} > 35$ min	n = 44 (51%)
Delta temperature A-B	$\Delta T > 4.7^\circ\text{C}$	n = 21 (24%)

Some potentially relevant references

Walcott, S. M., Kirkham, A. L., & Burns, J. M. (2020). Thermoregulatory costs in molting Antarctic Weddell seals: impacts of physiological and environmental conditions. *Conservation physiology*, 8(1), coaa022.

Sato, K., Tsuchiya, Y., Kudoh, S., & Naito, Y. (2003). Meteorological factors affecting the number of Weddell seals hauling-out on the ice during the molting season at Syowa Station, East Antarctica.

We thank the reviewer for the advised reference, which were very helpful to discuss our new results and to broaden the study to a larger range.

Reviewer #2 (Remarks to the Author):

Review of the manuscript "Breaking the fast: First report of dives and ingestions events in molting Southern elephant seals".

This paper present an original and extensive data set relying on the equipment of 55 southern elephant seal during the moult. These females were equipped with a time depth recorder, a stomach temperature pill and a satellite tag. Each female was monitored on average for 7.0 ± 2.9 days.

This species, is one of the three seal species experiencing a catastrophic moult (i.e. the renewal of both hair and part of the cornified epidermis. During the moult, elephant seals were considered as exclusively fasting extended period of fasting and in this species the moult period was previously thought to be spent entirely on land.

This study demonstrate are that, contrarily to what was previously assumed, 77% of the elephant seal females went swimming at sea and 39% dove.

Strength of the paper

The results are original, change the perception on elephant seal ecology-ecophysiology, and may have strong implication on studies implying to estimate metabolic rate by implementing tritium or deuterium on that species during the moult.

The results will be definitely of interest for the marine biologist and marine mammal community as well as researchers studying fasting metabolisms.

The sample size is significant and therefore the main findings are robust.

Main weaknesses

We thank the reviewer for the constructive comments regarding the strength and weaknesses of the manuscript. We answered below to the different points mentioned int the general comments.

Analyses could be developed regarding the influence of meteorological conditions, such as air temperature, wind, rain..., on the occurrence of such at-sea visit behaviour, were not investigated. I suppose such meteorological records must be available over the study period.

We thank the reviewer for suggesting this. We do have meteorological data available but the first plan was to separate the papers; one short note to report southern elephant seals dive and ingest during the molt, and one questioning the reasons for such behaviour. However, we agree it would be of interest to group both. We decided to add analyses of meteorological factors, added in the method section as follow : "Local meteorological data was retrieved from Météo-France base station archives in Port-aux-Français (Kerguelen archipelago). These data included temperature under cover (°C), relative humidity (%), insolation duration (min) and wind speed ($m.s^{-1}$). Data were recorded per hour and averaged per day [...] Generalized linear models were used to analyse the effect of temperature, relative humidity, insolation duration and wind speed on the probability female elephant seals probability to dive, surface swim and ingest. Models were fitted with a binomial distribution (logistic regressions for probabilities). Models were checked for overdispersion and adjusted with quasibinomial distribution if necessary."

Lines 196-200 and lines 229-233.

Results are shown in Table 5 (see below) and reported in the result section as follow (lines 285-295):

"The mean daily wind speed was 8.8 ± 2.7 m/s (ranging from 2.8 to 18.4 m/s), mean air temperature was 8.4 ± 2.4 °C (ranging from 4.4 to 15.3 °C), mean daily total sunshine duration was 355.3 ± 219.9 min (ranging from 0 to 855 minutes) and mean relative humidity was $70.2 \pm 7.7\%$ (ranging from 50.7 to 91.6%).

Diving behavior was not related to meteorological conditions (Table 5). However, we found a positive effect of temperature and a negative effect of wind speed on the probability of surface swimming (Table 5). Females tended to do more surface swimming during days of high temperature and days of low wind speed (Figures 5A and 5B).

We also found a positive effect of insolation on probability of ingestion. The probability of recording individuals that ingested increased during days with high total insolation (Figure 5C). The effect of relative humidity and wind speed on ingestion was not significant (Table 5)."

We added Figure 5 (see below) to show significant effects of daily windspeed, mean temperature and insolation.

Figure 5: Significant relationship between meteorological parameters and female southern elephant seals' behavior using. A: GLM predicted daily proportion of swimming females according to daily windspeed (m/s). B: GLM predicted daily proportion of swimming females according to daily mean temperature (°C). C: GLM predicted daily proportion of females that ingest according to daily mean insolation (min/day).

The diurnal pattern of at-sea visit was not investigated and this could provide valuable information about possible the factors driving such behaviour. Is this behaviour only taking place exclusively during the day, at night or throughout the whole day?

We thank the reviewer for this interesting suggestion. We looked at the diurnal pattern and noticed that 57% of the dives, 66% of the surface swimming, and 70% of the ingestion events occurred during the day.

I am not certain that mass in it-self is the most informative biometric factor. Depending on their sizes, seals with similar mass might be in fact in a very different state of body condition. Therefore, I would suggest the authors to perform the analyses on body condition changes rather than mass changes.

We thank the reviewer for pointing this and agree that changes in body mass do not represent changes in body condition in a reliable way. To improve these analyses, we calculated a body mass index (mass / size²) we included in our models to consider body condition at equipment. Also, when we investigated changes in body mass during the monitoring, rather than calculate a delta BMI, we looked at the delta mass but kept the length of the animal as a fixed variable in the model, as well as the monitoring length. Results of the models are available in Table 3 and 4. We plotted the results in Figure 4, as the daily specific body mass loss according to significant parameters in the model (PC1 as the diving score of the females, increasing when females dive often and for a long time (see method and result section lines 215-227 and 272-283), and the number of ingestions, see below).

Table 3. ANOVA table of the best linear mixed effect models for at-sea behavior (PC1 and PC2) and physiological consequences (body mass loss).

Group	Parameter	AICc	Explanatory variables	X ²	Df	P
Diving	PC1	63.3	Monitoring length	13.43	1	< 0.001 ***
Surface swimming	PC2	56.5	Monitoring length	0.34	1	0.56
Diving + Surface swimming	Body mass loss	126.7	Mass start	0.02	1	0.89
			Length	0.37	1	0.54
			PC1	8.19	1	0.004 **
			Number of ingestions	4.42	1	0.03 *
			Monitoring length	10.65	1	0.001 **
On land	Body mass loss	46.9	Mass start	0.38	1	0.53
			Length	0.27	1	0.61
			Monitoring length	1.41	1	0.24

Table 4. Summary of coefficients and goodness-of-fit indices from the best models for the number of ingestions (GLMER fitted with negative binomial law).

Group	Parameter	Explanatory variables	Coefficient \pm SE	Z	P
Diving + Surface swimming	Number of ingestions	Intercept	0.54 \pm 0.43	0.95	0.20
		Monitoring length	0.03 \pm 0.05	1.07	0.51
On land	Number of ingestions	Intercept	-1.95 \pm 2.95	-	0.51
		Monitoring length	0.29 \pm 0.48	0.60	0.55

Figure 4: Daily mass specific delta mass (g/kg/day) of female southern elephant seals according to : A. Their diving score (PC1 as individual values on the first component of the PCA) and B. The number of ingestions. Individuals with high PC1 values perform more dives, that last longer, and more surface swimming.

More importantly, the discussion of this paper remains very much elephant seal focussed. The general scope of this study needs to be broaden, regarding ecophysiological considerations such as peripheral perfusion and therefore fur synthesis. For instance, I got the impression than stomach temperature data are available both on land and at sea and, I would have found

interesting to compare both stomach temperature, excluding stomach temperature drops, both on lands and at sea.

We thank the reviewer for pointing this and agree with the interest to broaden the scope. We rewrote the introduction to highlight the importance of the molt regarding ecophysiological consideration, including other molting pinnipeds (lines 41-51 and 61-74).

We also broadened our discussion including other species and questioning the energetical constraints of molting, and how going at sea could impact energy reserves of the animals, molting speed, and what would be the consequences on individual's fitness (lines 409-441, 443-464).

We added on-land stomach temperature data, to detect hyperthermia events that could be responsible for at-sea behavior, due to heat stress. We reported the analyses in the method section (lines 176-180) and the results lines 255-260. We also compared at-sea and on land temperatures excluding temperature drops (see in the results section, lines 255-260).

It is mentioned in the paper that Argos location were used to determine if the stomach temperature drops were taking place on land or at sea. Clearly according to the precision of the Argos location categories (3,2, 1, 0...) it is, most of the time, impossible to distinguish using this criteria if seal were really on land or at sea as the seals remain very close to shore. So I guess only the combination of Argos locations but in combination with the wet and dry sensor information, could help to assess the land versus at-sea period.

At the end it is unclear to me if some of these temperature drops were observed while the seals were on land, and if it is the case this is likely to suggest water ingestion. This could be used to compare the temperature drop signal from these land events versus at-sea events and possibly to be used to further discriminate between possible water versus prey ingestion events. If not such events are detected on land this should be clearly stated.

We thank the reviewer for pointing this and agree this was not correctly stated in the result section. We changed the section as below :

"A summary of ingestion event variables is available in Table S1. We recorded a total of 87 ingestion events (Table 2), from 80% of individuals (n = 31 from total 39), with an average of 2.8 ± 2.1 ingestions per individual. Among these events, 70% occurred during the day. A total of 10 ingestion events were detected during a dive (12%) and 42 events occurred during surface swimming (49%). Due to bad Argos locations, the remaining 34 events (39%) were not clearly characterized as at sea or on land, but were not associated with complete submersion of the animal according to wet and light sensors. These ingestion events could have happened strictly on land or in the tidal zone.

Results of the different methods based on the literature⁴⁵ are reported in Table 2. Based on the different methods, 24 to 51% of the ingestions were classed as potential prey ingestions."

We thank the reviewer for suggesting to compare ingestion curves at sea vs. non-identified location to discriminate water vs. prey. However, the comparison did not show clear differences in ingestion curves, and was not sufficient to determine if it would salted water, water on land, or prey."

We also added detailed in the method section (lines 131-132):

"Because Argos locations were poor, diving patterns were analyzed only from depth, light level, and wet/dry sensor data."

My general feeling is that much more could be done with this exceptional dataset, and at the end, the overall study could be more convincing.

We thank the reviewer for the constructive comments and hope the manuscript to be more convincing regarding the supplementary analyses added, and the broader scope of the introduction and discussion.

Not being an English speaker myself, I cannot comment on the English grammar but I feel that the writing could be sharpened. I found many imprecisions throughout the document.

We thank the reviewer for pointing this. We rephrased many sentences and asked an English native speaker to read the manuscript before the new submission.

For these reasons, I recommend that this paper, if accepted, require to go through a major revision process.

Other comments:

1. Line 45 : angustirostris

We corrected line 63

2. Line 63: However, our data revealed at-sea movements and ingestion events that match with Boyd et al. hypothesis. This hypothesis should be clearly formulated here and not much later in the text. i.e. that prey and/water ingestion was likely to take place during the molt.

The Boyd hypothesis refers to previous suggestions of water ingestion (1993) based on metabolic calculations. We deleted the term "Boyd hypothesis" and introduced earlier in the text as follow; line 78-81: ". Although water ingestion has been suggested as a mechanism for SES to mitigate physiological costs while molting³⁴, there have been few studies examining movement and potential drinking or feeding behavior in this species throughout the molt."

3. Line 123 :

Light-level variations due to the day-night cycle were removed using a seasonal decomposition function based on moving averages (function "seasonal decompose" from Python package "statsmodels" version 0.13.2.

This is unclear to me why this was done. Is this to distinguish between day and night?

We thank the reviewer for pointing this. We clarified in the text line 134-136 as follows:

"To make sure light level variations involved depth and not daily changes, day-night light-level variations were removed using a seasonal decomposition function based on moving averages (function "seasonal decompose" from Python package "statsmodels"; version 0.13.2)."

4. Line 144 : Again it is unclear to me how and why the light level was used in this paper.

We used the light level to confirm animals were diving. We compiled data from depth, wet/dry and light sensors, to make sure the animal is not in a mud pool or on its back. We rephrased as follows:

Lines 134-139 : To make sure light level variations involved depth and not daily changes, day-night light-level variations were removed using a seasonal decomposition function based on moving averages (function "seasonal decompose" from Python package "statsmodels"; version 0.13.2). We combined a depth reaching threshold of 0.5 m and a "wet" display to spot the start of a dive. To distinguish mud pools from sea-water baths, we made sure the light-level varied by more than 5 units (values ranging from 0 to 255) during the presumed dive."

Lines 171-172 : "Six animals with less than 2 days of recording were discarded as seasonal decomposition of light-level function requires two cycles of 24 hours to apply."

5. Line 156 : A visual check of stomach temperature curves over time for each ingestion led to the removal of anomalous events

How this was decided or defined as an anomalous event, this appear to be very suggestive.

We thank the reviewer for pointing this and rephrased as follow lines 158-160:

“Following the algorithm analysis, a visual check of ingestion curves led to the removal of 12 anomalous events (non-logarithmic curves or abrupt variations consistent with STP measurement errors).”

6. Line 157 : “leading to an estimated rate of only 11% of false positives detected by our algorithm”.

This should go in the result part. Above you mention a visual check and here an algorithm. This part remain very unclear to me.

We thank the reviewer for noticing this was unclear. Ingestion events were detected using an algorithm, and after that, a visual check was made to confirm the values and shape of the curves were not related to STP measurements errors. We rephrased the section as follow (lines150-162): “Ingestion events were spotted by a sharp decline in stomach temperature followed by a logarithmic increase (Figure 2). The algorithm looked for (i) the minimum stomach temperature value below 36.5°C (point B, Figure 2A, based on core temperatures already measured in northern and SES); and then (2)the previous temperature drop (point A, Figure 2A) and point at which temperature returned to baseline within 10 minutes ($\pm 0.1^\circ\text{C}$, point C, Figure 2A). Events with a too-short temperature recovery duration between points B and C (<20 s, based on our own data exploration) were considered inconsistent with ingestions and deleted. Multiple close events were merged and considered as one. Every ingestion event was classified according to the behavior of the female (on land, diving, or surface swimming). Following the algorithm analysis, a visual check of ingestion curves led to the removal of 12 anomalous events (non-logarithmic curves or abrupt variations consistent with STP measurement errors).”

7.How much drinking took place why seals were on land.

Due to bad Argos location when seals remain on-land, we are not sure if some ingestions happen on land or at sea. Although we compiled data of the wet and light sensor, we could not discriminate if the ingestion took place on land or in the tidal zone, with no complete submersion of the animal, i.e if they involved fresh or salted water. We clarified lines 265-268:

“A total of 10 ingestion events were detected during a dive (12%) and 42 events occurred during a surface swimming (49%). Due to bad Argos locations, the remaining 34 events (39%) were not clearly characterized as at-sea or on-land, but were not associated with complete submersion of the animal according to wet and light sensor. These ingestion events could have happened strictly on land or in the tidal zone.”

8. Line 232, 234, 241: for the first time,. This study is the first, It is the first time...

A bit too much for me, once should be enough!

We rephrased as follow

Line 298-300: “Through the use of time-depth recorders and stomach temperature loggers, we were able to identify surface swimming, diving, and ingestion events during the molt of female SES. To our knowledge, this study is the first to show evidence of SES breaking the fast during this energetically costly molt period.”

Line 311 : “To our knowledge, this behavior has never been reported in a species displaying a “catastrophic” molt” “

We deleted the others “first” in the discussion.

9. Line 246 to 248. This would have been better in the introduction than in the discussion. We thank the reviewer and added this in the introduction to highlight the energetic constraints of the molt in marine mammals, lines 53-56 : "In phocids, the maintenance of a warm skin for extended periods for the shedding and renewal of hair is required as tissue growth ceases below 17°C, which is inconsistent with permanent cold water exposure where heat is lost 25 times faster than in air. Maintaining a high surface temperature in a cold environment results in high metabolic costs; with molting metabolic rate estimated to increase by a factor of 2-3 compared to resting metabolic rate."

10. Line 283-286 : Unclear to me how this was performed We thank the reviewer for pointing this and added more details in the method section, lines 182-192 :

"To distinguish between water and prey ingestions, we followed two published methods, using the index of the rate of stomach temperature recovery (I , $s \cdot ^\circ C^{-1}$), calculated as follow:

$$I = t_{0.5} / (T_1 - T_2)$$

where T_1 ($^\circ C$) is the initial stomach temperature before the drop due to the ingestion of sea water, T_2 ($^\circ C$) is the minimal temperature reached during the drinking or feeding event, and $t_{0.5}$ (s) the time from the beginning of the temperature decrease to the point of half recovery (half-way temperature between initial A and minimal B; Figure 2). The threshold of $250 s \cdot ^\circ C^{-1}$ was used to distinguish between fish and water consumption, as described by Kuhn and Costa, 2006.

We also calculated the area above the curve (integral) of drop in stomach temperature during ingestion events, as it was supposed to be related to ingested mass in northern elephant seals, based on the trapezoid method (for more details, see Figure S2 in Supplementary information)."

We also simplified Table 2 to focus on the different methods to estimate the number of feeding event (see below), without considering where they occurred.

Table 2: Summary of ingestion type based on Kuhn and Costa (2006) results, from 31 female southern elephant seals monitored between 2014 and 2022 (n = 87 events).

Method	Threshold for feeding events	Number of feeding events
Index of rate of temperature recovery I ($s \cdot ^\circ C^{-1}$)	$I \text{ value} > 250 s \cdot ^\circ C^{-1}$	n = 37 (43%)
Area above the curve ($s \cdot ^\circ C$)	Integral $> 3000 s \cdot ^\circ C$	n = 24 (28%)
Time of temperature recovery (Δt between B-C)	$t_{rec} > 35 \text{ min}$	n = 44 (51%)
Delta temperature A-B	$\Delta T > 4.7^\circ C$	n = 21 (24%)

11. The quality of the graph and figures require improvements.

We thank the reviewers and improved the figures.

Figure 1 : we rephrased the legend to give a better definition of a dive, dive cycle, surface swimming, and deleted the bottom graph as it did not bring anything new.

Figure 2 : we changed the numbers in letters (A,B,C,D) for the different panels, and changed A-D to present the "classic ingestion curve" before showing the others. We also replaced temperature

points A,B and C by T_A , T_B , T_C to make it easier to read.

Figure 3, 4 and 5 were deleted and replaced by new analyses illustrations.

Figure 3: Principal Component Analysis (PCA) including number of dives per individual, number of surface swims per individual, and mean dive or surface swimming duration. The first component (PC1) accounted for 59.0% of the variation and the second component (PC2) accounted for 23.8% of the variation. PC1 was characterized by number of dives, number of surface swims, and mean dive duration. PC2 was defined by surface swimming duration. Individuals are reported on the PCA and identified through their behavior (red, diving; green, on land; blue, surface swimming only). Green dot represents eight individuals on a same point.

Figure 4: Daily mass loss per unit of body mass (g/kg/day) of female southern elephant seals according to: A. Their diving score (PC1 as individual values on the first component of the PCA) and B. The number of ingestions per individual. Individuals with high PC1 values perform more dives, that last longer, and show more surface swimming.

Figure 5: Significant relationship between meteorological parameters and female southern elephant seals' behavior using. A: GLM predicted daily proportion of swimming females according to daily windspeed (m/s). B: GLM predicted daily proportion of swimming females according to daily mean temperature (°C). C: GLM predicted daily proportion of females that ingest according to daily mean insolation (min/day).

Fig S1 : we modified the scale with a black vertical line to make it easier to read.

At-sea movement recorded in one monitored individual at mid-stage of moult in 2016, from the site of Pointe Suzanne (49°26'S, 70°26'E), with a corresponding dive maximal depth of 18.5 m.

Grey dots represent Argos locations and red arrows represent the direction of movements between the dots.

Reviewer #3 (Remarks to the Author):

This manuscript represents a considerable and significant piece of work showing for the first time the evidence of breaking the fast during the energetically costly period of catastrophic molting for a marine mammal.

This work provides unique and critical cues on the ingestion events of Southern elephant seals and it was very interesting to see how the instruments were used to detect those diving and ingestion events. This research changes our perception of the molt for such predator and withdraw the hypothesis of complete fasting. This research is of immediate relevance for people in our field but also for a broader audience, as it discusses unexpected behavior and behavioral changes during a key period of the life cycle that could be interesting for both terrestrial and marine studies.

These observed ingestion events could provide valuable insights into the physiology of pinnipeds regarding their hydration during such costly period of their life cycle and their foraging behavior, very different from the rest of their diving behavior outside of the molt. It also represents an exceptional dataset for sub-antarctic pinniped species.

Analysis sounds robust, correct, and conducted in the right way. In the introduction, I would appreciate the authors extend on other marine seabird and mammals in terms of their molting strategy. The authors discussed it in the discussion section but more background in the introduction on other species would be greatly appreciated. In the methods section, the approach to differentiate water from prey ingestions should be clearly stated. In the results, I would appreciate adding a map of all tracking data covering the full period and coloring differently prey ingestion from water ingestion. The part that needs the most improvements is the discussion. I suggest providing 1) perspectives in order to improve the study to better quantify change in body mass, differences between water ingestion and prey ingestion; 2) a general discussion about potential preys found at such depths and their quality; 3) hypotheses regarding the consequences of such behavior on individual fitness, breeding success, and the role of inter-individual variability in such behavior; 4) the role of hydration for sub and Antarctic pinnipeds; 5) finally, I strongly suggest a short paragraph discussing the effect of temperature pills on the animal welfare. I understand it is not often easy to quantify the effect, but I recommend at least to provide some hypothesis on the potential effect of such pills.

I appreciated a very detailed description of the methods and the relevance of the figures.

Following those comments, I highly recommend this paper to be published with the suggested minor revisions. Please find below the minor edits requested on top of my general comments.

We thank the Reviewer 3 for the nice and constructive comments about the manuscript. We considered the suggestions concerning the introduction and the discussion and agree that it would be of interest to more readers if both are written in a broader context. We decided to rewrite the introduction focusing on the molt, and it's energetic constraints. We added examples of other molting species and pinnipeds such as Weddell seals, harbor seals, spotted seals, ringed seals. We also rewrote the discussion adding other species, focusing in the reasons and consequences of going at sea during the molt. We added perspectives to go further in ingestion analyses (the use of sonar tag, already used to detect the type of preys in SES), discussed what type of prey would be available and what would be the effect of such behavior on individual's

fitness. As we added new analyses based on meteorological data, we discussed the effect of thermal stress in pinnipeds and the potential need for hydrate while remaining on-land. We also added a short paragraph on possible adverse effects of stomach temperature pills and why we believe these effects would be minors in our study (short retention time, small STP size compared to stomach volume..).

Abstract

1- Unless this is a journal requirement, I would suggest the authors adding just a little quantitative information about the percentage of time spent feeding and ingesting water during the whole molting period.

We rewrote the abstract adding the suggested information as follows (lines 20-29):

“Southern elephant seals (SES) experience a ‘catastrophic molt’, a costly event characterized by the renewal of both hair and epidermis that requires high peripheral vascular circulation. During the molt animals are therefore constrained by high metabolic heat loss and are thought to fast and remain on land.

To examine the ability of individuals to balance the energetic constraints of molting on land we investigated the stomach temperature and movement patterns of molting female SES.

We found that 79% of females swam and 61% ingested water or prey items, despite the cost of cold-water exposure while molting. This behavior was related to periods of warm and calm conditions, and females that dived and ingested more often lost less body mass.

We conclude that the paradigm of fasting during the molt in this species, and the fitness consequences of this behavior should be reconsidered, especially in the context of a changing climate.”

Introduction

2- L45-46: could the authors please put the latin names in parenthesis just after the species name.

We corrected as suggested line 61-64: “Southern elephant seals (SES) are one of only four species of phocids that undergo a ‘catastrophic molt’, the only one in a polar environment, during which they renew both their hair and cornified epidermis (with northern elephant seal (NES), *Mirounga angustirostris*, Hawaiian monk seal (*Monachus schauinslandi*), and Mediterranean monk seal (*Monachus monachus*)”

Material and methods

3- L86: could the authors please provide details about the retrieval of the temperature pills when not excreted?

We added details in the method section lines 116-118: “Tags were retrieved at recapture, while remaining STP were naturally ejected later at sea or on land. Individuals were recaptured between the completion of the molt and their final departure to sea.”

The STP do not remain in the stomach more than 22 days in captivity (Kuhn and Costa 2006).

They are always spit out at sea or naturally ejected.

4- Figure S2: I suggest using a different color for the dots, for separating the 45% with estimated error radius <250 m from the other ones.

We than the reviewer for the suggestion. We decided to delete Fig S2 because of few Argos locations (short monitoring) and bad quality locations, as animals are often on land, on the back,

or in the mud. We kept Fig S1 to show a good quality at-sea travel recording but we did not use other Argos locations. At-sea travels were determined using depth, wet and light level captors.

5- L90: "9 of which were successfully associated with a STP for more than 2 days", could the authors please clarify, I am unsure of what the authors meant here with associated for more than 2 days?

We thank the reviewer for pointing this as it was not clear; we meant we could analyze data of more than 2 days because the seasonal decomposition function requires 2 days to apply (to distinguish "day-night" light level variations from "depth" light level variations). We rewrote as follow (lines 119-121):

"Of the 55 instrumented seals, we obtained stomach temperature data over 2 days from 39 of the seals. Data were analyzed starting from 6 hours following capture (to discard the effects of anesthesia), until the STP was ejected, or the seal recaptured"

And lines 171-172 : "Six animals with less than 2 days of recording were discarded as seasonal decomposition of light-level function requires two cycles of 24 hours to be valid."

6- L95: the reference is not in the same format as other references.

We thank the reviewer for noticing this and changed the format of the reference.

7- L138: "We then filtered out events that were too short in duration to be ingestions", how did the authors decide the threshold for duration?

A threshold of 20 s was decided based on our own data exploration; data were recorded every 10 s and mean recovery time was 36.8 +/- 17.7 min. We clarified lines 154-156 as follow: "Events with a too-short temperature recovery duration between points B and C (<20 s, based on our own data exploration) were considered inconsistent with ingestions and deleted."

8- L144: please replace "those 6" by "those 6 individuals"

We rephrased the section lines 171-172 as follow: "Six animals with less than 2 days of recording were discarded as seasonal decomposition of light-level function requires two cycles of 24 h to be valid."

9- L148: please replace "down sample" by "down-sampled"

We rephrased this sentence as follow, line 163-165: "To examine STP temperature drift over time we downsampled the data to one measurement every 15 min (factor of 100) and examine the trend in temperature over time using a Kendall's tau correlation coefficient."

10- L185: it is not in Table S2 but figure S2?

We thank the reviewer for pointing this mistake in the manuscript. We deleted this figure as explained in comment number 4 (see above).

Results

11- It would be great to have a figure with all tracking data combined and colours would be the individuals

We thank the reviewer for the suggestion. However, as we had few good quality Argos locations and many females located at the same place (Pointe Suzanne or Estacade), we are afraid it would be confusing to add a global with few and low precision locations data. We kept Fig S1 to show a good quality at-sea travel recording.

12- Figure 4: why are they dots with the exact same value for at-sea events? Did the authors round the results?

We agree with the reviewer that this figure is unclear and does not revealed statistical analyses. We deleted this one and reported the molting stage at equipment in the result section lines 273-275: "Molt stage when animals were equipped was $54 \pm 25\%$ with 41% of the females caught at the initial stage (n = 16), 41% at the mid stage (n = 16), and 18% at the final stage of molt (n = 7)."

We also included the molting stage at equipment in the models to see if it impacted at-sea travel or ingestions. This parameter was not included in the best model so we changed Fig4 (now showing the daily mass specific delta mass (g/kg/day) of female southern elephant seals according to their diving score and the number of ingestions).

13- L213: are these results significant? please specify whether it is or not. If not, the authors should be careful and say "the number of ... seems/tends to increase..."

We agree with the reviewer and deleted this figure. Results were not significant but did not take into account the monitoring length, size of the individuals, molting stage...

We explained in the method section lines 222-224: "We used LMM to test the effect of diving, surface swimming (PC1, PC2), ingestion counts, parameters at the beginning of monitoring (body mass at capture, body length, molting stage), and monitoring duration on body mass loss (kg)."

And in the result section lines 280-283 : "For females going to sea (group diving + group surface swimming), we found a positive relationship with PC1 and of the number of ingestions on daily body mass loss (Table 3). Females that dived and ingested more often lost less mass per unit of body mass (Figures 4A and 4B)."

Table 3. Anova table of the best linear mixed effect models for at-sea behavior (PC1 and PC2) and physiological consequences (body mass loss).

Group	Parameter	AICc	Explanatory variables	X²	Df	P
Diving	PC1	63.3	Monitoring length	13.43	1	< 0.001 ***
Surface swimming	PC2	56.5	Monitoring length	0.34	1	0.56
Diving + Surface swimming	Body mass loss	126.7	Mass start	0.02	1	0.89
			Length	0.37	1	0.54
			PC1	8.19	1	0.004 **
			Number of ingestions	4.42	1	0.03 *
			Monitoring length	10.65	1	0.001 **
On land	Body mass loss	46.9	Mass start	0.38	1	0.53
			Length	0.27	1	0.61
			Monitoring length	1.41	1	0.24

Figure 4: Daily mass loss per unit of body mass (g/kg/day) of female southern elephant seals according to: A. Their diving score (PC1 as individual values on the first component of the PCA) and B. The number of ingestions per individual. Individuals with high PC1 values perform more dives, that last longer, and show more surface swimming.

14- Figure 5: the legend mentions number of panels while the figure has letters, please correct this is confusing.

We changed Figure 5 which now illustrates the effect of meteorological parameters on females' behavior (see below).

Figure 5: Significant relationship between meteorological parameters and female southern elephant seals' behavior using. A: GLM predicted daily proportion of swimming females according to daily windspeed (m/s). B: GLM predicted daily proportion of swimming females according to daily mean temperature (°C). C: GLM predicted daily proportion of females that ingest according to daily mean insolation (min/day).

15- L217-227: it is unclear how the authors differentiate from prey ingestion from water ingestion, I may have missed something, but this should be clarified and clearer.

We thank the reviewer for pointing this. We clarified in the methods lines 182-192 :

"Ingestion events

To distinguish between water and prey ingestions, we followed two published methods, using the index of the rate of stomach temperature recovery (I , $s \cdot ^\circ C^{-1}$), calculated as follow:

$$I = t_{0.5} / (T_1 - T_2)$$

where T_1 ($^\circ C$) is the initial stomach temperature before the drop due to the ingestion of sea water, T_2 ($^\circ C$) is the minimal temperature reached during the drinking or feeding event, and $t_{0.5}$ (s) the time from the beginning of the temperature decrease to the point of half recovery (half-way temperature between initial A and minimal B; Figure 2). The threshold of $250 s \cdot ^\circ C^{-1}$ was used to distinguish between fish and water consumption, as described by Kuhn and Costa, 2006.

We also calculated the area above the curve (integral) of drop in stomach temperature during ingestion events, as it was supposed to be related to ingested mass in northern elephant seals, based on the trapezoid method (for more details, see Figure S2 in Supplementary material)."

Discussion

16- L253: please add "and" in between "molt" and "also".

We rephrased lines 318-320 as follow: "Therefore, our results reveal that SES may be the first recorded case of a species experiencing a catastrophic molt while also spending time at sea, despite the potential energetic costs."

17- L259-261: I suggest the authors to add some hypothesis regarding the prey they may target at such depths and their energetic quality.

We thank the reviewer for the suggestion and added a paragraph in the third part of the discussion when discussing effect on body condition, lines 427-441.

18- L266-271: given the data you have and the figure 3, I would not discuss any difference that are neither visible nor significant. I would comment more on the small sample size limiting the study and on the perspectives of what the authors should do to test this.

We agreed with the reviewer and deleted this part as we changed the analyses. Now the difference in body mass loss is discussed line 441-441 (see response to comment 17).

19- L292-295: this information is very useful and should be clearly written in the methods section. We reported the information in the method section line 188-189 : "The threshold of 250 s.°C⁻¹ was used to distinguish between fish and water consumption, as described by Kuhn and Costa, 2006."

20- L246: please change the wording "why".

We rephrased the discussion as follow lines 312-320: "In marine mammals, pinnipeds such as elephant seals undergoing such an extreme physiological event, have always been thought to rely exclusively on energy reserves in blubber during the molt. All birds experiencing catastrophic molt are known to fast and avoid cold water contact, as demonstrated in cormorants and penguins. Few diving birds are known to break the fast and dive during the molting period, but they undergo a rapid simultaneous molt that is not reported as 'catastrophic' (common eider *Somateria mollissima*, lesser snow geese *Chen caerulescens caerulescens*, and grebes *Podiceps sp.*). Therefore, our results reveal that SES may be the first recorded case of a species experiencing a catastrophic molt while also spending time at sea, despite the potential energetic costs."

21- L308-310: please clarify, this is confusing, I did not get the message.

We thank the reviewer for pointing this. We rewrote the discussion and clarified this part lines 418-426: "Moreover, going to sea during the molt might not be as costly as we believed. Daily specific body mass loss of our females was a good indicator of energetic loss during the molt. A previous study reported that the mass loss of female elephant seals averages 4.7 kg.d⁻¹, during 25 to 30 days of molting. Surprisingly, females that went to sea did not lose more body mass than females remaining on land but had higher stomach temperatures, which could reflect a higher metabolic rate, hence higher energy loss at sea. However, previous studies on molting Arctic seals (ringed *Pusa hispida*, spotted *Phoca largha*, and bearded seals *Erignathus barbatus*) reported that metabolic rates were similar when resting in air and water. An increase in stomach temperature might be detected at the onset of diving as a result of heat generated by muscles during swimming, which questions the reliability of STP to estimate metabolic variation."

22- L307-311: I suggest adding more words discussing the potential threshold of fat depletion that will lead to the decision to go at-sea. Moreover, it is not clear in the paragraph whether it is the need for water ingestion or feeding that motivate breaking the fast. I would also improve the discussion by adding a paragraph on the way the change in body fat could be monitored and improved, in order to identify those thresholds, very important to understand the behavior the authors are describing.

We thank the reviewer for the suggestion. As we rewrote the discussion, the third part is now focused on possible physiological need to go at sea to rebuild fat reserves (lines 409-441). We ended the paragraph suggesting a more accurate method to investigate the type of prey (lines 439-441), which would be the easiest thing to investigate: "However, no precise information on what was ingested could be determined with this method, and thus this question requires further investigation. Other methods could bring important information such as sonar tag or video camera which would allow to know more precisely what is ingested during these trips to sea during the molt".

The use of deuterium labelled water could also help to follow more precisely changes in body composition and fat reserves, but the application in the field and during this period (need to remain close to the shelter, females' movements...), we believe it would be difficult to have these data.

23- L319-320: did the authors tested for this? I recommend testing it with surface temperatures on departure at-sea compared with the average at this period.

We thank the reviewer for the suggestions. Surface temperature were not available data in this project but we got meteorological data we added to the analyses to investigate the effect of thermal stress. Also, we looked at hyperthermia events through STP recordings to see if increasing stomach temperature could motivate at-sea travel. We reported this in the method section lines 176-180 : "We investigated on-land stomach temperatures to detect hyperthermia events that could have initiated at-sea movements. We calculated the mean stomach temperature ($36.6 \pm 0.2^{\circ}\text{C}$) and considered hyperthermia if values were upper than the 97.5% quantile for more than one hour. We looked if hyperthermia events were followed by at-sea movements."

And the results section lines 255-260 : "Hyperthermia events (from 1 to 3 events) were detected for 61.5% of the animals ($n = 24$), with a mean stomach temperature of $38.1 \pm 0.3^{\circ}\text{C}$. In total, 33 events of 110 ± 39 min were analyzed, and the first at-sea event occurred within 24.5 ± 21.0 hours after."

24- L321: please remove "good"

We rewrote the discussion as we added new analyses; now the effect of meteorological parameters is discussed in the second paragraph lines 350-407. We changed the term "good" or "bad" weather as follow, lines 356-357: "Our data revealed that during days with low wind speed and days with high air temperature, female SES tend to swim more often. The same parameters were reported to influence the number of molting Weddell seals (*Leptonychotes weddellii*) hauling out on ice, but surprisingly, it was the opposite behavior, as Weddell seals tend to remain in the water when wind speed is elevated and air temperature are low."

25- I suggest adding a paragraph at the end on hypothesis regarding the fitness consequences, breeding success...of such behavior (breaking the fast)

We thank the reviewer for this suggestion. We added a paragraph at the end of the discussion lines 443 to 464 to discuss the possible effects of changes in molt phenology or energetic constraints.

Sara Labrousse

Reviewers' comments:

Reviewer #1 (Remarks to the Author):

In this study, the authors document for the first time that many elephant seals undergoing a catastrophic molt spend time in the water and ingest prey and/or water. They do a nice job identifying why the seals may spend time in the water and what some of the benefits may be. They also suggest future research areas to better understand the drivers of this behavior.

The authors did a wonderful job including additional analyses that were suggested during the 1st round of reviews. The writing is also stronger, particularly the discussion. While the writing throughout is improved, the manuscript still needs some significant editing of the Intro, Methods and Results section (just minor edits in the discussion). I point out some errors, unclear sentences, and where flow can be improved and offer some suggestions. Most of my suggestions are in the introductions and methods. By the results section, I was limited on time and could not edit language anymore, but I point out some of the places where I got confused and where the writing needs improvement to ensure the reader understands what they did and can interpret the results.

Specific comments

Pg 2 Line 41: suggest replace "... integuments and influenced ..." with "... integuments, which is influenced ..."

Pg 2 line 44: sentence does not make sense as written, made a suggest that made grammatically correct, but still a little unclear = suggest: "...fitness, as a change," but still unclear

Pg 2 line 47: replace "requiring" with "a" - Don't need to say again since you already use word requiring earlier in sentence.

Pg 2 line 49 & 56: use of "phocid seals" - this is redundant - just use one or the other - okay to define what a phocid is at first use but then would just use one from that point on.

Pg 2 line 49-60: Paragraph should be more clear and concise - - would be stronger and more concise if not in passive voice and sentences broken up a bit differently: here is a suggestion (needs refinement) to try to make this paragraph stronger based on what I think your main point is. "... Regrowth of skin and fur requires a skin temperature above 17C. Maintenance of this warm temperature in polar regions leads to elevated metabolic rates. To reduce energy expenditure during the molt, seals are insulated with thick subcutaneous blubber, regulate peripheral blood flow, and avoid even higher heat loss by hauling out to minimize time in the cold water, where heat loss is 25 times faster than in air. Even with these actions, molting metabolic rate is 2-3 higher than resting metabolic rate. ..."

Pg 2 Line 61-62: A little unclear. I think clarity can be improved by breaking into 2 sentences? Something like: "... a 'catastrophic molt', during which they renew both their hair and cornified epidermis." Of the four species, they are the only one to undergo the molt in a polar environment (other species:....)"

Pg 2, third paragraph: Flow should be improved. The authors jump from conditions on land, to why they think they haul out, and then back to conditions on land. I would try to rewrite so maybe start with mass change and increased peripheral circulation that combined would result in high heat loss so that is why they should haul out and fast. But even if they are on land they experience challenging conditions and then on to how environmental conditions can influence heat loss. The current flow makes it a little hard for the reader to know exactly where you are going.

Pg 2, Line 64-66: Try to reduce passive voice throughout. This will improve clarity and conciseness. SES molt on subantarctic islands, where they experience cold, wet, and windy conditions during the ~1-month haulout.

Pg 3, Fourth paragraph: this paragraph is jumpy with few clear connections between sentences. For example, the jump to water ingestion is a bit abrupt and unclear why this has been suggested as a

mechanism to reduce costs. An explanation and a better connection to the first few sentences is needed. There is missing cohesion between thoughts/sentences in this paragraph.

Pg 3, last paragraph can be stronger: For example Line 91 – ‘we wanted to find out’ is a bit too casual for a manuscript. Example: In this study, we investigated the behavior of adult female SES during the molt. Females were instrumented with TDRs to document diving behavior and stomach temperature pills to document ingestion events. We hypothesized that females would remain on land, either in mud pools or on the beach, fasting, at least during the early stages of the molt to minimize heat loss. If seals did spend time at sea, we investigated if weather conditions influenced this behavior. Last we hypothesized that only females in good body condition would spend time at sea, as they have sufficient energy reserves to compensate for the high heat loss.

Pg 4: Line 138: At sea – should be “At-sea”

Pg 5 – Fig 1 – I know you have the y axis scaled so you can see the variability, but to show how they are different behaviors, it would be nice to have depth and light level y-axis be the same for all of them (ex. Depth 0 -32 and light 100-~170?. Then you don’t need to fit the tick numbers in between A and B and C and D (they were cut off for some).

Methods

Pg 6 Lines 157-162. Writing is unclear and a little challenging to understand. I think you can label the figure better to match the words and the words can be more direct. I assume you are trying to say that you used an algorithm to identify the 3 variables you describe. As written, I first thought this is what the algorithm was programmed to look for to identify a feeding event, but that does not make sense for ii) and iii). Below is an example of a clearer way to state what I think you are trying to say. I unfortunately did not quite understand what you meant in ii) about the drop in temp. I thought it meant the change from start to minimum, but then you refer to T1 – but that is the starting value. If it is the drop, then I would move T2 to the where you have a change in Temp and the T1 to the minimum temp (where you have T2). I also did not understand what you meant by returning to baseline within 10 minutes. It looks like it usually takes longer than 10 minutes. I could not figure out what you mean using the figure. If my interpretation below is incorrect, it still needs revising so the reader can easily understand what you are doing.

For each sharp decline, the algorithm identified i) the minimum temperature below 36.5C (average core body temperature of elephant seals)(T1 on Fig 2A), ii) the drop in temperature between the starting value and the minimum value (T2), and iii) the point at which the temperature returned to baseline (T3, Fig 2). Events that took <20 s to recover (time between minimum temperature and recovery temperature) were deleted because they were too short to represent an ingestion. Events that were close together (DO YOU HAVE A TIME CRITERIA?) were merged into a single event. Each ingestion event was assigned to a female behavior: on land, diving, surface swimming, or unknown.

Pg 6, Line 176 – unclear to me why the seasonal decomposition of light level requires two cycles. I am not sure I understand what you mean by seasonal light decomposition.

Figure 2 – line 192-193, there is repetition of the definition of T0.5

Pg 7, Line 200-201 – the last sentence is very repetitive of the first. I don’t think you need this unless you add details about how what method you used to look at the relationship between hyperthermia and at-sea movements. Otherwise, it just repeats the first sentence.

Pg 7 – suggest flipping the order of hyperthermia and ingestion events. Ingestion events is more related to the previous section and would flow better if came before hyperthermia.

Pg 8, line 225 – replace “have been” with “were” and change Results were to “Results are”

Pg 8 – statistics section – would help the reader if you explain why you did things. Why did you do a PCA analysis – was that how you determined the swimming score or did you do it with the hope you would get groups of individuals? If looking for groups, why? Try to tie things back specifically to your objectives/hypotheses

Pg 9, line 254 – unclear what you mean by ‘monitoring length’ as written (Applies to Table 3 and 4 too)? Suggested using duration instead of length (unless you are talking about length during the study period). I would also delete group from PC1 and PC2 clarification – that made me think of your animal

groupings but that is not what the PC's are saying.

Pg 10, line 280 – don't need to say between two dives of the same cycle (you already defined this).

Pg 10, line 283 – In table 1 you say 31 females are listed, including 15 that dove. But here you say 30 females surface swam, including those 15. What did the last seal do? Or is there a typo here or in the table?

Pg 11 – Table 1 maximal dive depth should be maximum dive depth

Pg 13, Table 3 – I am having difficulty understanding what you did. I don't understand why PC1 and PC2 are parameters in the first two models and then an explanatory variable in the 3rd. You need to better explain how you broke up these models in your stats section. One thing that might help is if you very clearly tied back your analysis to your hypotheses in the intro and introduced your methods in the same order you present your 3 objectives and then present your results in the same order. Things were a bit mixed up, making it harder to follow. In your stats section, you can state things like: To determine if weather influenced when seals spent time in the water we ran a LMM with x as a fixed effect, y and a dependent variable, and z as a random effect. I was confused by the groups and what the models were trying to answer in this table and had to work too hard to figure it out. The flow in the discussion was great because it matched your intro – would be good to organize the stats and results section to match.

Pg 15, Table 5 – The parameter name of % of diving females and % of swimming females is slightly confusing. I don't understand what the response variable is by this name. Needs a better name so the reader easily knows what this is. Is this the probability that a female will go diving in certain weather conditions? Or from figure 5 it sounds like the proportion of females. Was the response variable the percent of the tagged females swimming during an hour with certain weather conditions. I am sure if I go back and read things more carefully, I can figure this out, but it needs to be easier for the reader to figure out.

Reviewers' comments:

Reviewer #1 (Remarks to the Author):

In this study, the authors document for the first time that many elephant seals undergoing a catastrophic molt spend time in the water and ingest prey and/or water. They do a nice job identifying why the seals may spend time in the water and what some of the benefits may be. They also suggest future research areas to better understand the drivers of this behavior.

The authors did a wonderful job including additional analyses that were suggested during the 1st round of reviews. The writing is also stronger, particularly the discussion. While the writing throughout is improved, the manuscript still needs some significant editing of the Intro, Methods and Results section (just minor edits in the discussion). I point out some errors, unclear sentences, and where flow can be improved and offer some suggestions. Most of my suggestions are in the introductions and methods. By the results section, I was limited on time and could not edit language anymore, but I point out some of the places where I got confused and where the writing needs improvement to ensure the reader understands what they did and can interpret the results.

We wish to thank Reviewer 1 for her/his kind words on the manuscript, constructive comments that have helped us to improve the paper, and for taking the time to make so many suggestions and rephrasing many parts of the paper. We followed all the suggestions concerning the writing, and we reorganized the result section so it fits better to the methods. We hope improvement now ensure the reader to easily understand and interpret the results. We also read carefully the discussion and improved the flow.

Pg 2 Line 41: suggest replace "... integuments and influenced ..." with "... integuments, which is influenced ..."

We changed as suggested line 41

Pg 2 line 44: sentence does not make sense as written, made a suggest that made grammatically correct, but still a little unclear = suggest: "...fitness, as a change," but still unclear

We changed the sentence line 44 as followed: "Molting is involved with a change in coat or plumage color, that may be related to reproductive success and camouflage in the environment^{5,11}, growth¹⁰, thermoregulation¹², and defense against parasites or pathogens⁷, hence playing an important role in the fitness of an individual."

Pg 2 line 47: replace "requiring" with "a" - Don't need to say again since you already use word requiring earlier in sentence.

We corrected as suggested line 48

Pg 2 line 49 & 56: use of "phocid seals" – this is redundant – just use one or the other – okay to define what a phocid is at first use but then would just use one from that point on.

We corrected as suggested; we kept Phocid seals at first use line 49 but put "phocid" only line 56

Pg 2 line 49-60: Paragraph should be more clear and concise - – would be stronger and more concise if not in passive voice and sentences broken up a bit differently: here is a suggestion (needs refinement) to try to make this paragraph stronger based on what I think your main point is. "... Regrowth of skin and fur requires a skin temperature above 17C. Maintenance of this warm temperature in polar regions leads to elevated metabolic rates. To reduce energy expenditure during the molt, seals are insulated with thick subcutaneous blubber, regulate peripheral blood flow, and avoid even higher heat loss by hauling out to minimize time in the cold water, where heat loss is 25 times faster than in air. Even with these actions, molting metabolic rate is 2-3 higher than resting metabolic rate. ..."

We thank the reviewer for the concise suggestion. We rephrased as suggested lines 51-55.

Pg 2 Line 61-62: A little unclear. I think clarity can be improved by breaking into 2 sentences? Something like: "... a 'catastrophic molt', during which they renew both their hair and cornified epidermis." Of the four species, they are the only one to undergo the molt in a polar environment (other species:....)"

We changed as suggested lines 59-61.

Pg 2, third paragraph: Flow should be improved. The authors jump from conditions on land, to why they think they haul out, and then back to conditions on land. I would try to rewrite so maybe start with mass change and increased peripheral circulation that combined would result in high heat loss so that is why they should haul out and fast. But even if they are on land they experience challenging conditions and then on to how environmental conditions can influence head loss. The current flow makes it a little hard for the reader to know exactly where you are going.

We thank the reviewer for the suggestion and reorganized the paragraph as advised, on line 62-71:

"During this period, like other phocids, they lose a large proportion of their body mass (23% SES, 25%, NES, 17%, gray seals *Halicheorus grypus*). This large change, combined with the increased peripheral circulation which would result in high heat loss in water, led to the conclusion that elephant seals fast and remain on land during the molt. While ashore, they are exposed to a range of meteorological conditions. As SES molt on subantarctic islands, they experience cold, wet, and windy conditions during the approximately one-month haulout³⁴. Environmental conditions strongly influence heat flux between an animal and the environment. SES minimize heat loss through social thermoregulation in large aggregations and through choice of different beach habitats such as grass or mud pools. Therefore, we would predict that only animals with sufficient energy reserves to cover the cost of fasting, molting, and cold-water contact would be able to go to sea during the molting period."

Pg 2, Line 64-66: Try to reduce passive voice throughout. This will improve clarity and conciseness. SES molt on subantarctic islands, where they experience cold, wet, and windy conditions during the ~1-month haulout.

We rephrased as suggested lines 67-68.

Pg 3, Fourth paragraph: this paragraph is jumpy with few clear connections between sentences. For example, the jump to water ingestion is a bit abrupt and unclear why this has been suggested as a mechanism to reduce costs. An explanation and a better connection to the first few sentences is needed. There is missing cohesion between thoughts/sentences in this paragraph. We thank the reviewer for pointing this. We connected better the sentences and added an explanation why ingestions could reduce the cost of molting. See lines 72-85:

"A high energetic cost during the molt increases the depletion of energy reserves in blubber, with possible fitness consequences, such as lower survival or reproductive success for seals. Climate in the Southern Ocean is rapidly changing, and environmental conditions are known to impact heat loss and molting phenology in phocids. In this context, understanding how energetic constraints and individual strategies may help reduce metabolic costs of animals during key stages of their lifecycle is important. While it is commonly agreed that molting SES remain on land and rely on their reserves, breaking the fast has already been suggested as a mechanism to mitigate physiological costs while molting by restoring blubber reserves or limiting metabolic water production. However, few studies examined movement and potential drinking or feeding behavior in this species throughout the molt. One way to investigate at-sea feeding or dinking behavior is to combine stomach temperature measurements with time-depth recorders (TDRs). This allows detection of possible cold water or prey item ingestion through stomach temperature changes and to record the depth and location of ingestion events. This approach has previously been used in seabird research (penguins, cormorants, albatrosses, and captive or non-captive pinnipeds (NES and Californian sea lions), but not during the molt."

Pg 3, last paragraph can be stronger: For example Line 91 – 'we wanted to find out' is a bit too casual for a manuscript. Example: In this study, we investigated the behavior of adult female SES during the molt. Females were instrumented with TDRs to document diving behavior and stomach temperature pills to document ingestion events. We hypothesized that females would remain on land, either in mud pools or on the beach, fasting, at least during the early stages of the molt to minimize heat loss. If seals did spend time at sea, we investigated if weather conditions influenced this behavior. Last we hypothesized that only females in good body condition would spend time at sea, as they have sufficient energy reserves to compensate for the high heat loss. We thank the reviewer for the suggestion and rephrased as suggested lines 86-92.

Pg 4: Line 138: At sea – should be "At-sea"
We corrected as suggested line 140.

Pg 5 – Fig 1 – I know you have the y axis scaled so you can see the variability, but to show how they are different behaviors, it would be nice to have depth and light level y-axis be the same for all of them (ex. Depth 0 -32 and light 100-~170?. Then you don't need to fit the tick numbers in between A and B and C and D (they were cut off for some).

We thank the reviewer for the suggestion. We modified the figure as advised as follow lines 149-154:

Figure 1: Diving parameters during different at-sea behaviors; diving or surface swimming. Surface swimming occurs between 1 and 5 m, while a dive is characterized by a maximum depth of more than 5 m. A dive or a surface swimming cycle is characterized by at least two successive dives or surface swimming separated by less than 20 min at the surface. Fig 1A. represents a surface swimming cycle and different profiles of dive cycles (B, C, D) of a female southern elephant seal in 2017. Each dive is defined by the succession of a submerged phase (red area) and a surface phase (green areas) (B).

Methods

Pg 6 Lines 157-162. Writing is unclear and a little challenging to understand. I think you can label the figure better to match the words and the words can be more direct. I assume you are trying to say that you used an algorithm to identify the 3 variables you describe. As written, I first thought this is what the algorithm was programmed to look for to identify a feeding event, but that does not make sense for ii) and iii). Below is an example of a clearer way to state what I think you are trying to say. I unfortunately did not quite understand what you meant in ii) about the drop in temp. I thought it meant the change from start to minimum, but then you refer to T1 – but that is the starting value. If it is the drop, then I would move T2 to the where you have a change in Temp and the T1 to the minimum temp (where you have T2). I also did not understand what you meant

by returning to baseline within 10 minutes. It looks like it usually takes longer than 10 minutes. I could not figure out what you mean using the figure. If my interpretation below is incorrect, it still needs revising so the reader can easily understand what you are doing.

For each sharp decline, the algorithm identified i) the minimum temperature below 36.5C (average core body temperature of elephant seals)(T1 on Fig 2A), ii) the drop in temperature between the starting value and the minimum value (T2), and iii) the point at which the temperature returned to baseline (T3, Fig 2). Events that took <20 s to recover (time between minimum temperature and recovery temperature) were deleted because they were too short to represent an ingestion. Events that were close together (DO YOU HAVE A TIME CRITERIA?) were merged into a single event. Each ingestion event was assigned to a female behavior: on land, diving, surface swimming, or unknown.

We thank the reviewer for the suggestion and changed the writing; however, the order of i) and ii) is the opposite; the algorithm first looks for the minimum temperature (T2) and then it searches for the previous stable temperature (the starting one, T1) .We clarified as follow lines 158-170:

“Ingestion events were identified by a sharp decline in stomach temperature followed by a logarithmic increase (Figure 2). The ingestion detection algorithm identified:

1. The minimum temperature below 36.5°C (under the lowest core temperature described for elephant seals^{36,44}), referred to as T₂
2. The previous stable temperature before the drop (T₁, backward from T₂)
3. The point at which temperature returned to baseline for at least 10 minutes^{44,47} (T₃, forward from T₂)

Events that were close together were merged into a single event. Each ingestion event was assigned to a female behavior: on land, diving, surface swimming, or unknown. Following the algorithm analysis, a visual check of ingestion curves led to the removal of 12 anomalous events (non-logarithmic curves or abrupt variations consistent with STP measurement errors). The STP was considered lost when the stomach temperature reading remained constant for at least 3 hours. Outliers were deleted using a filter based on the rolling average and bounded by the 1st and 3rd quartiles. T₁, T₂ and T₃ are illustrated in figure 2A.”

Pg 6, Line 176 – unclear to me why the seasonal decomposition of light level requires two cycles. I am not sure I understand what you mean by seasonal light decomposition.

We apologize if the explanation was not clearer the second time. The seasonal decomposition of light level allows to “erase” natural light variations that are registered by the logger. After this filter, we only have the light variations related to depth changes (and then at-sea evidence). But to apply on our data, this function needs a record of minimum 2 days to “recognize” the daily pattern of changes in light level, and filter our data with this “recognized daily pattern”. Here the link to the documentation we added in the manuscript:

https://www.statsmodels.org/stable/generated/statsmodels.tsa.seasonal.seasonal_decompose.html

We clarified lines 183-187:

STP were synchronized with diving patterns, that were filtered with a seasonal decomposition of light-level function to cancel the effect of the day/night cycle. As this function required 48 hours to work, six animals with less than 2 days of recording were discarded (R documentation). Eight

seals ejected the pill before recapture. In total, 80% of monitored seals were recaptured with the temperature pill still recording (n = 31), with an average retention duration of 7.2 ± 2.9 days.”

Figure 2 – line 192-193, there is repetition of the definition of T_{0.5}

We thank the reviewer for noticing this and corrected in the legend (as noticed, we made a confusion between T_{0.5} and t_{0.5}; T_{0.5} is the half-way recovery temperature between T₁ and T₂, while t_{0.5} is the time of half-way recovery). We rectified line 193-195.

Pg 7, Line 200-201 – the last sentence is very repetitive of the first. I don't think you need this unless you add details about how what method you used to look at the relationship between hyperthermia and at-sea movements. Otherwise, it just repeats the first sentence.

We thank the reviewer for the suggestion and deleted the last sentence to avoid repetition (line 212).

Pg 7 – suggest flipping the order of hyperthermia and ingestion events. Ingestion events is more related to the previous section and would flow better if came before hyperthermia.

We thank the reviewer for the suggestion and flipped the order as advised lines 199-217.

Pg 8, line 225 – replace “have been” with “were” and change Results were to “Results are”
We corrected as suggested line 219.

Pg 8 – statistics section – would help the reader if you explain why you did things. Why did you do a PCA analysis – was that how you determined the swimming score or did you do it with the hope you would get groups of individuals? If looking for groups, why? Try to tie things back specifically to your objectives/hypotheses

We thank the reviewer for pointing this. At first we wanted to keep all individuals together in the analyses with one variable, “going to sea”. However, some individuals dove, others did surface swimming, more or less frequently, with long or short bouts... to define a real “swimming score”, we did a PCA. The score would be easier to include in the models, to see what would be the impact on BM loss of a high swimming score for example. However, when we plotted the individuals, we noticed we had two groups, each more related to one axis or the other. In that context, we kept PC1 and PC2 as a swimming score, and separated the groups according to their behavior when group size was sufficient.

Pg 9, line 254 – unclear what you mean by ‘monitoring length’ as written (Applies to Table 3 and 4 too)? Suggested using duration instead of length (unless you are talking about length during the study period). I would also delete group from PC1 and PC2 clarification – that made me think of your animal groupings but that is not what the PC's are saying.

We changed “length” by “duration” line 252 and in Table 3 and 4 as suggested.

We agree with the reviewer that the writing is confusing; we deleted the groups from PC1 and PC2 and changed by : “We decided to define three groups according to the behavior of each female (diving, surface swimming, on land, Figure 3) in the models when group size was sufficient.” Lines 227-229.

Pg 10, line 280 – don't need to say between two dives of the same cycle (you already defined this).

We corrected as suggested line 285.

Pg 10, line 283 – In table 1 you say 31 females are listed, including 15 that dove. But here you say 30 females surface swam, including those 15. What did the last seal do? Or is there a typo here or in the table?

We thank the reviewer for pointing this. To be more precise, 14 seals dived and performed surface swimming. For the last one who dived, we could not detect surface swimming. We clarified lines 308-310: "Seventy-seven percent of monitored females (n = 30) performed surface swimming. Fourteen animals dived and surface swam. One female only dived."

Pg 11 – Table 1 maximal dive depth should be maximum dive depth

We corrected as suggested in Table 1

Pg 13, Table 3 – I am having difficulty understanding what you did. I don't understand why PC1 and PC2 are parameters in the first two models and then an explanatory variable in the 3rd. You need to better explain how you broke up these models in your stats section. One thing that might help is if you very clearly tied back your analysis to your hypotheses in the intro and introduced your methods in the same order you present your 3 objectives and then present your results in the same order. Things were a bit mixed up, making it harder to follow. In your stats section, you can state things like: To determine if weather influenced when seals spent time in the water we ran a LMM with x as a fixed effect, y and a dependent variable, and z as a random effect. I was confused by the groups and what the models were trying to answer in this table and had to work too hard to figure it out. The flow in the discussion was great because it matched your intro – would be good to organize the stats and results section to match.

We thank the reviewer for pointing this and reorganized the stat section. We started with how we investigated the reasons for variability in SES behavior, then the effect of meteorological factors, and finally what would be the consequences of swimming / ingesting in SES conditions before leaving for their next trip. We rewrote line 340-401.

In that context, we changed Tables 3-4-5 and figures 4-5.

Table 3 contains models to explain SES behavior variability. Table 4 shows best models for environmental conditions effect on females' behavior, and figure 4 the results. Table 5 shows the best models to explain females' condition at the end of the mold, and figure 5 the results.

Table 3. best models to explain female SES behavior variability (**A.** Swimming score and **B.** Ingestion behavior). LMM were used for diving and surface swimming behavior (PC1 and PC2). A generalized mixed linear model was used for ingestion behavior (number of ingestions).

A. ANOVA table of the best linear mixed effect models for at-sea behavior (PC1 and PC2)

Response variable	Behavior / group	Explanatory variables	X ²	Df	P
-------------------	------------------	-----------------------	----------------	----	---

PC1	Diving	Monitoring duration	13.43	1	< 0.001 ***
PC2	Surface swimming	Monitoring duration	0.34	1	0.56

B. Summary of coefficients and goodness-of-fit indices from the best models for the number of ingestions (GLMER fitted with negative binomial law).

Response variable	Location	Explanatory variables	Coefficient ± SE	Z	P
Number of ingestions	At-sea	Intercept	0.54 ± 0.43	0.95	0.20
		Monitoring duration	0.03 ± 0.05	1.07	0.51
	On land	Intercept	-1.95 ± 2.95	-0.66	0.51
		Monitoring duration	0.29 ± 0.48	0.60	0.55

Table 4. Summary of coefficients and goodness-of-fit indices from the best models for environmental conditions effect on diving behavior, swimming behavior, number of ingestions. GLMS were fitted with binomial (probability of ingesting) or quasibinomial (probability of diving and surface swimming) law.

Response variable	Explanatory variables	Coefficient ± SE	t	P
Probability of diving	Intercept	-1.7 ± 0.8	-2.0	0.04 *
	Temperature	0.06 ± 0.08	0.8	0.44
	Wind speed	-0.09 ± 0.06	-1.4	0.16
Probability of surface swimming	Intercept	-1.6 ± 0.6	-2.7	0.008 **
	Temperature	0.21 ± 0.06	3.7	<0.001 ***
	Wind speed	-0.10 ± 0.05	-2.1	0.03 *
Probability of ingesting	Intercept	-3.7 ± 1.7	-2.1	0.03 *
	Sunlight	0.001 ± 0.0006	2.8	0.02 *
	Humidity	0.03 ± 0.02	1.7	0.09
	Wind speed	-0.09 ± 0.05	-1.8	0.07

Figure 4: Significant relationships between meteorological conditions and behavior of female southern elephant seals using. A: GLM predicted daily proportion of swimming females with daily mean windspeed (m/s). B: GLM predicted daily proportion of swimming females with daily mean temperature (°C). C: GLM predicted daily proportion of females that ingest with total daily sunshine duration (min/day).

Table 5. ANOVA table of the best linear mixed effect models for female SES condition at the end of the molting period (body mass loss), grouped by location (on land or at sea).

Response variable	Group	AICc	Explanatory variables	χ^2	Df	P
Body mass loss	On land	46.9	Mass start	0.38	1	0.53
			Length	0.27	1	0.61
			Monitoring duration	1.41	1	0.24
	At-sea	126.7	Mass start	0.02	1	0.89
			Length	0.37	1	0.54
			PC1	8.19	1	0.004 **
			Number of ingestions	4.42	1	0.03 *
			Monitoring duration	10.65	1	0.001 **

Figure 5: Daily mass loss per unit of body mass (g/kg/day) of female southern elephant seals according to: A. Their diving score (PC1 as individual values on the first component of the PCA) and B. The number of ingestions per individual. Individuals with high PC1 values perform more dives, that last longer, and show more surface swimming.

Pg 15, Table 5 – The parameter name of % of diving females and % of swimming females is slightly confusing. I don't understand what the response variable is by this name. Needs a better name so the reader easily knows what this is. Is this the probability that a female will go diving in certain weather conditions? Of from figure 5 it sounds like the proportion of females. Was the response variable the percent of the tagged females swimming during an hour with certain weather conditions. I am sure if I go back and read things more carefully, I can figure this out, but it needs to be easier for the reader to figure out.

We thank the reviewer for the suggestion. We corrected with “proportion of diving / surface swimming / ingesting females” in Table 5.